# Decoding the physics of observed actions in the human brain

**Moritz F Wurm[1]\*, Doruk Yiğit Erigüç[1,2]**

[1]CIMeC – Center for Mind/Brain Sciences, University of Trento, Rovereto, Italy; [2]Max Planck Institute for Human Cognitive and Brain Sciences, Leipzig, Germany

## eLife Assessment

In an **important** fMRI study with an elegant experimental design and rigorous cross-decoding analyses, this work shows a **convincing** dissociation between two parietal regions in visually processing actions. Specifically, aIPL is found to be sensitive to the causal effects of observed actions, while SPL is sensitive to the patterns of body motion involved in those actions. The work will be of broad interest to cognitive neuroscientists, particularly vision and action researchers.

**Abstract** Recognizing goal-directed actions is a computationally challenging task, requiring not only the visual analysis of body movements, but also analysis of how these movements causally impact, and thereby induce a change in, those objects targeted by an action. We tested the hypothesis that the analysis of body movements and the effects they induce relies on distinct neural representations in superior and anterior inferior parietal lobe (SPL and aIPL). In four fMRI sessions, participants observed videos of actions (e.g. breaking stick, squashing plastic bottle) along with corresponding point-light-display (PLD) stick figures, pantomimes, and abstract animations of agent–object interactions (e.g. dividing or compressing a circle). Cross-decoding between actions and animations revealed that aIPL encodes abstract representations of action effect structures independent of motion and object identity. By contrast, cross-decoding between actions and PLDs revealed that SPL is disproportionally tuned to body movements independent of visible interactions with objects. Lateral occipitotemporal cortex (LOTC) was sensitive to both action effects and body movements. These results demonstrate that parietal cortex and LOTC are tuned to physical action features, such as how body parts move in space relative to each other and how body parts interact with objects to induce a change (e.g. in position or shape/configuration). The high level of abstraction revealed by cross-decoding suggests a general neural code supporting mechanical reasoning about how entities interact with, and have effects on, each other.

**\*For correspondence:**
moritz.wurm@unitn.it

**Competing interest:** The authors declare that no competing interests exist.

## Introduction

Action recognition is central for navigating social environments, as it provides the basis for understanding others' intentions, predicting future events, and social interaction. Many actions aim to induce a change in the world, often targeting inanimate objects (e.g. opening or closing a door) or persons (e.g. kissing or hitting someone). Recognizing such goal-directed actions is a computationally challenging task, as it requires not only the temporospatial processing of body movements, but also processing of how the body interacts with, and thereby induces an effect on, the object targeted by the action, for example a change in location, shape, or state. While a large body of work has investigated the neural processing of observed body movements as such (**Grossman et al., 2000**; **Giese and Poggio, 2003**; **Puce and Perrett, 2003**; **Peuskens et al., 2005**; **Peelen et al., 2006**), the neural

**Figure 1.** Schematic illustration of action effect structures and cross-decoding approach. (**A**) Simplified schematic illustration of the action effect structure of 'opening'. Action effect structures encode the specific interplay of temporospatial object relations that are characteristic for an action type independently of the concrete object (e.g. a state change from closed to open). (**B**) Cross-decoding approach to isolate representations of action effect structures and body movements. Action effect structure representations were isolated by training a classifier to discriminate neural activation patterns associated with actions (e.g. 'breaking a stick') and testing the classifier on its ability to discriminate activation patterns associated with corresponding abstract action animations (e.g. 'dividing'). Body movement representations were isolated by testing the classifier trained with actions on activation patterns of corresponding point-light-display (PLD) stick figures.

mechanisms underlying the analysis of action effects, and how the representations of body movements and action effects differ from each other, remain unexplored.

The recognition of action effects builds on a complex analysis of spatial and temporal relations between entities. For example, recognizing a given action as 'opening a door' requires the analysis of how different objects or object parts (e.g. door and doorframe) spatially relate to each other and how these spatial relations change over time. The specific interplay of temporospatial relations is usually characteristic for an action type (e.g. opening, as opposed to closing), independent of the concrete target object (e.g. door or trash bin), and is referred to here as *action effect structure* (*Figure 1A*). In addition, action effects are often independent of specific body movements – for example, we can open a door by pushing or by pulling the handle, depending on which side of the door we are standing on. This suggests that body movements and the effects they induce might be at least partially processed independently from each other. In this study, we define action effects as induced by intentional agents, but the notion of action effect structures might be generalizable to physical changes as such (e.g. an object's change of location or configuration, independently of whether the change is induced by an agent or not). Moreover, we argue that representations of action effect structures are distinct of conceptual action representations: The former capture the temporospatial structure of an object change (e.g. the separation of a closing object element), the latter capture the meaning of an action (e.g. bringing an object into an opened state to make something accessible) and can also be activated via language (e.g. by reading 'she opens the box'). Previous research suggests that conceptual action knowledge is represented in left anterior lateral occipitotemporal cortex (LOTC) (*Watson et al., 2013*; *Lingnau and Downing, 2015*; *Wurm and Caramazza, 2022*) whereas structural representations of action effects have not been investigated yet.

We argue that object- and movement-general representations of action effect structures are necessary for the recognition of goal-directed actions as they allow for inferring the induced effect (e.g. that something is opened) independently of specific, including novel, objects. Strikingly, humans recognize actions even in the absence of any object- and body-related information: In the animations of *Heider and Simmel, 1944*, the only available cues are abstract geometrical shapes and how these shapes move relative to each other and to scene elements. Yet, humans automatically and effortlessly attribute actions to these animations (e.g. opening, chasing, hiding, kissing), which argues against an inferential process and rather points toward an evolutionary optimized mechanism in the service of action recognition. Here, we test for the existence of a processing stage in the action recognition hierarchy that encode action effect representations independently from representations of body movements. We argue that both the recognition of body movements and the effects they induce rely critically on distinct but complementary subregions in parietal cortex, which is associated with visuospatial processing (*Goodale and Milner, 1992*; *Kravitz et al., 2011*), action recognition (*Caspers*

*et al., 2010*), and mechanical reasoning about manipulable objects (*Binkofski and Buxbaum, 2013*; *Leshinskaya et al., 2020*) and physical events (*Fischer et al., 2016*; *Fischer and Mahon, 2021*). Specifically, we hypothesize that the neural analysis of action effects relies on anterior inferior parietal lobe (aIPL), whereas the analysis of body movements relies on superior parietal lobe (SPL). aIPL shows a representational profile that seems ideal for the processing of action effect structures at a high level of generality: Action representations in bilateral aIPL generalize across perceptually variable action exemplars, such as opening a bottle or a box (*Wurm and Lingnau, 2015*; *Hafri et al., 2017*; *Vannuscorps et al., 2019*), as well as structurally similar actions and object events, for example, a girl kicking a chair and a ball bouncing against a chair (*Karakose-Akbiyik et al., 2023*). Moreover, aIPL is critical for understanding how tools can be used to manipulate objects (*Goldenberg and Spatt, 2009*; *Reynaud et al., 2016*). More generally, aIPL belongs to a network important for physical inferences of how objects move and impact each other (*Fischer et al., 2016*).

Also the recognition of body movements builds on visuospatial and temporal processing, but their representation should be more specific for certain movement trajectories (e.g. pulling the arm toward the body, regardless of the movement's intent to open or close a door). The visual processing of body movements has been shown to rely on posterior superior temporal sulcus (*Grossman et al., 2000*; *Giese and Poggio, 2003*; *Puce and Perrett, 2003*; *Peuskens et al., 2005*; *Peelen et al., 2006*). However, recent research found that also SPL, but less so aIPL, encodes observed body movements: SPL is more sensitive in discriminating actions (e.g. a girl kicking a chair) than structurally similar object events (e.g. a ball bouncing against a chair) (*Karakose-Akbiyik et al., 2023*). Moreover, point-light-displays (PLDs) of actions, which convey only motion-related action information but not the interactions between the body and other entities, can be decoded with higher accuracy in SPL compared to aIPL (*Yargholi et al., 2023*). Together, these findings support the hypothesis of distinct neural systems for the processing of observed body movements in SPL and the effect they induce in aIPL.

Using an fMRI-based cross-decoding approach (*Figure 1B*), we isolated the neural substrates for the recognition of action effects and body movements in parietal cortex. Specifically, we demonstrate that aIPL encodes abstract representations of action effect structures independently of motion and object identity, whereas SPL is more tuned to body movements irrespective of visible effects on objects. Moreover, cross-decoding between pantomimes and animations revealed that right aIPL represents action effects even in response to implied object interactions. These findings elucidate the neural basis of understanding the physics of actions, which is a key stage in the processing hierarchy of action recognition.

## Results

To isolate neural representations of action effect structures and body movements from observed actions, we used a cross-decoding approach: In four separate fMRI sessions, right-handed participants observed videos of actions (e.g. breaking a stick, squashing a plastic bottle) along with corresponding PLD stick figures, pantomimes, and abstract animations of agent–object interactions (*Figure 2*) while performing a simple catch-trial-detection task (see Methods for details).

To identify neural representations of action effect structures, we trained a classifier to discriminate the neural activation patterns associated with the action videos, and then tested the classifier on its ability to discriminate the neural activation patterns associated with the animations (and vice versa). We thereby isolated the component that is shared between the naturalistic actions and the animations – the perceptually invariant action effect structure – irrespective of other action features, such as motion, object identity, and action-specific semantic information (e.g. the specific meaning of 'breaking a stick').

Likewise, to isolate representations of body movements independently of the effect they have on target objects, we trained a classifier on action videos and tested it on the PLD stick figures (and vice versa). We thereby isolated the component that is shared between the naturalistic actions and the PLD stick figures – the coarse body movement patterns – irrespective of action features related to the target object, such as the way they are grasped and manipulated, and the effect induced by the action.

Additionally, we used pantomimes of the actions, which are perceptually richer than the PLD stick figures and provide more fine-grained information about hand posture and movements. Thus, pantomimes allow inferring how an object is grasped and manipulated. Using cross-decoding between

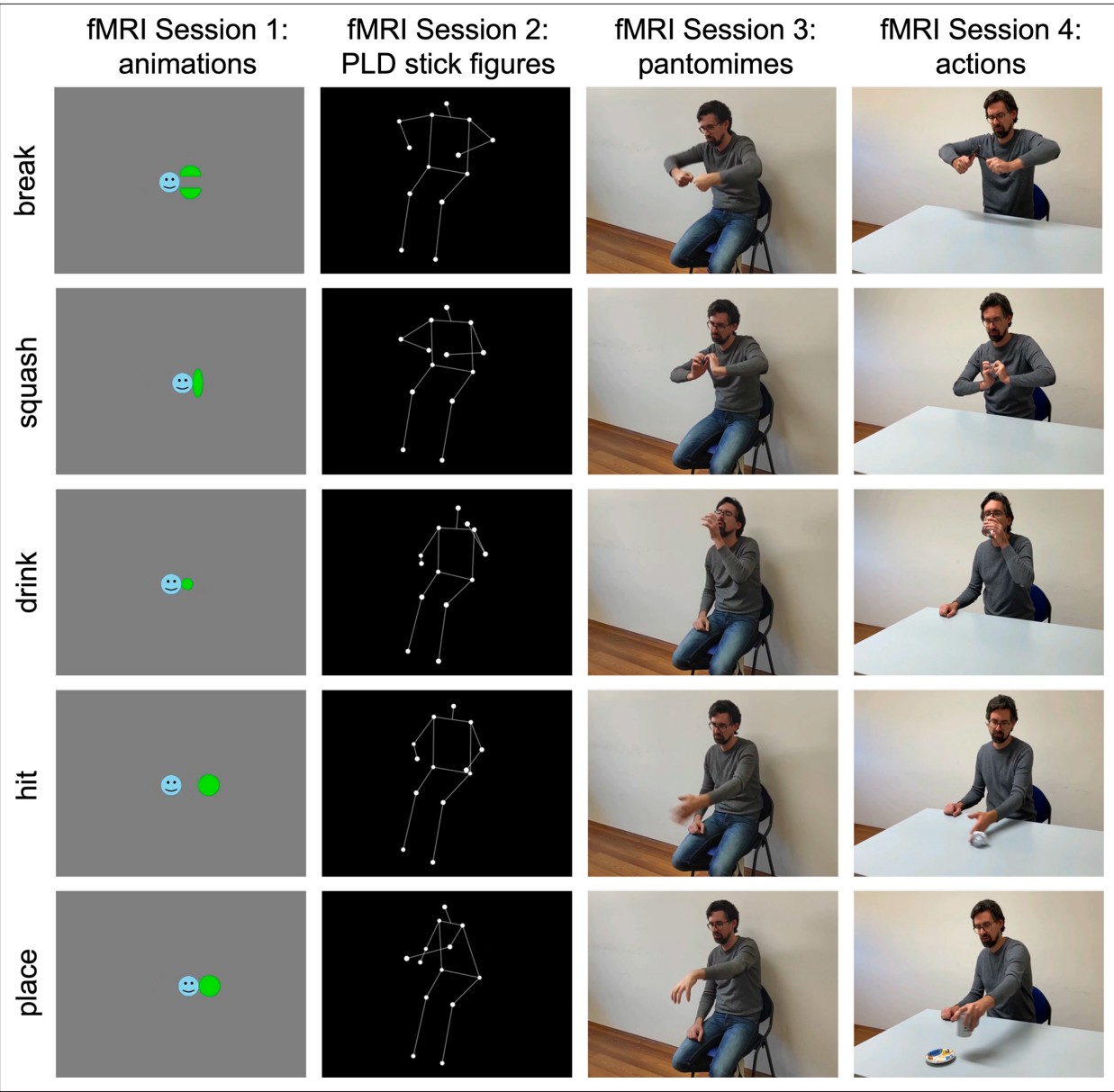

**Figure 2.** Experimental design. In four fMRI sessions, participants observed 2-s-long videos of five actions and corresponding animations, point-light-display (PLD) stick figures, and pantomimes. For each stimulus type, eight perceptually variable exemplars were used (e.g. different geometric shapes, persons, viewing angles, and left–right flipped versions of the videos). A fixed order of sessions from abstract animations to naturalistic actions was used to minimize memory and imagery effects.

pantomimes and animations, we tested whether action effect representations are sensitive to implied hand–object interactions or require a visible object change.

## Cross-decoding of action effect structures and body movements

We first tested whether aIPL is more sensitive in discriminating abstract representations of action effect structures, whereas SPL is more sensitive to body movements. Action-animation cross-decoding revealed significant decoding accuracies above chance in left aIPL but not left SPL, as well as in right aIPL and, to a lesser extent, in right SPL (*Figure 3A*). Action-PLD cross-decoding revealed the opposite pattern of results, that is, significant accuracies in SPL and, to a lesser extent, in aIPL. A repeated measures ANOVA with the factors region of interest (ROI; aIPL, SPL), TEST (action-animation, action-PLD), and HEMISPHERE (left, right) revealed a significant interaction between ROI and TEST ($F(1,23) = 35.03$, p = 4.9E−06), confirming the hypothesis that aIPL is more sensitive to effect structures

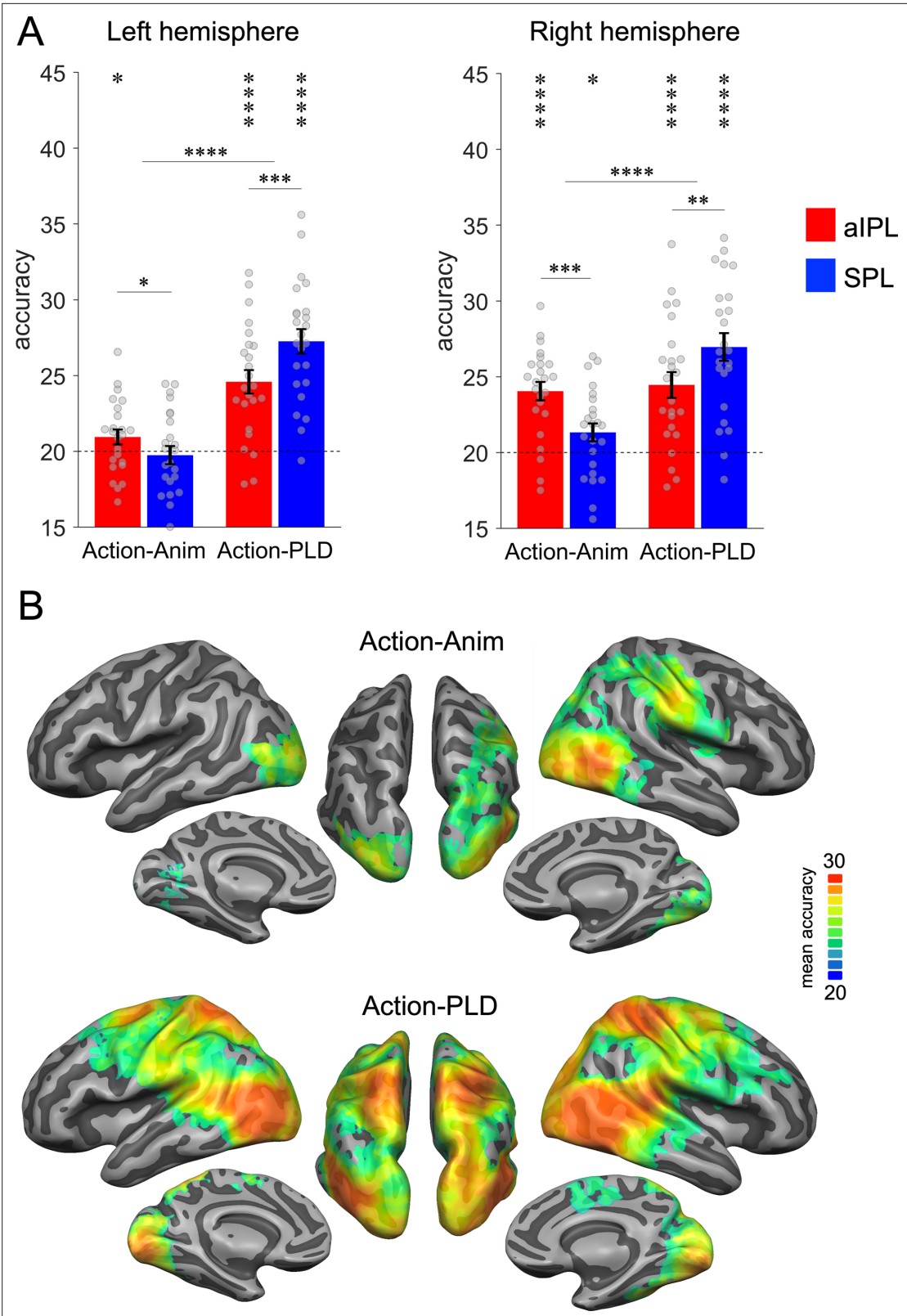

**Figure 3.** Cross-decoding of action effect structures (action-animation) and body movements (action-PLD). (**A**) Region of interest (ROI) analysis in left and right anterior inferior parietal lobe (aIPL) and superior parietal lobe (SPL) (Brodmann areas 40 and 7, respectively; see Methods for details). Decoding of action effect structures (action-animation cross-decoding) is stronger in aIPL than in SPL, whereas decoding of body movements (action-PLD cross-decoding) is stronger in SPL than in aIPL. Asterisks indicate FDR-corrected significant decoding accuracies above chance (*p < 0.05, **p <

*Figure 3 continued on next page*

*Figure 3 continued*

0.01, ***p < 0.001, ****p < 0.0001). Error bars indicate SEM (N=24). (**B**) Mean accuracy whole-brain maps thresholded using Monte-Carlo correction for multiple comparisons (voxel threshold p = 0.001, corrected cluster threshold p = 0.05). Action-animation cross-decoding is stronger in the right hemisphere and reveals additional representations of action effect structures in lateral occipitotemporal cortex (LOTC).

The online version of this article includes the following figure supplement(s) for figure 3:

**Figure supplement 1.** Cross-decoding of action effect structures (action-animation) and body movements (action-PLD) in left and right LOTC, pSTS, and V1 (respectively; see Methods for details on region of interest [ROI] definition).

of actions, whereas SPL is more sensitive to body movements. Post hoc *t*-tests revealed that, for action-animation cross-decoding in both left and right hemispheres, decoding accuracies were higher in aIPL than in SPL (left: $t(23)$ = 1.81, p = 0.042, right: 4.01, p = 0.0003, one-tailed), whereas the opposite effects were found for action-PLD cross-decoding (left: $t(23)$ = −4.17, p = 0.0002, right: −2.93, p = 0.0038, one-tailed). Moreover, we found ANOVA main effects of TEST ($F(1,23)$ = 33.08, p = 7.4E−06), indicating stronger decoding for action-PLD versus action-animation cross-decoding, and of HEMISPHERE ($F(1,23)$ = 12.75, p = 0.0016), indicating stronger decoding for right versus left ROIs. An interaction between TEST and HEMISPHERE indicated that action-animation cross-decoding was disproportionally stronger in the right versus left hemisphere ($F(1,23)$ = 9.94, p = 0.0044).

These findings were corroborated by the results of a searchlight analysis (*Figure 3B*): Within the parietal cortex, action-animation cross-decoding revealed a cluster peaking in right aIPL, whereas the parietal clusters for the action-PLD cross-decoding peaked in bilateral SPL. The whole-brain results further demonstrated overall stronger decoding for action-PLD throughout the action observation network, and in particular in LOTC extending into pSTS (see also *Figure 3—figure supplement 1* for ROI results in LOTC and pSTS), which was expected because of the similarity of movement kinematics between the naturalistic actions and the PLDs. Note that we were not interested in the representation of movement kinematics in the action observation network as such, but in testing the specific hypothesis that SPL is disproportionally sensitive to movement kinematics as opposed to aIPL. Interestingly, the action-animation cross-decoding searchlight analysis revealed an additional prominent cluster in right LOTC (and to a lesser extent in left LOTC), suggesting that not only aIPL is critical for the representation of effect structures, but also right LOTC. We therefore include LOTC in the following analyses and discussion. In addition, we observed subtle but significant above chance decoding for action-animation in bilateral early visual cortex (EVC; see also *Figure 3—figure supplement 1*). This was surprising because the different stimulus types (action videos and animations) should not share low-level visual features. However, it is possible that there were coincidental similarities between action videos and animations that were picked up by the classifier. To assess this possibility, we tested whether the five actions can be cross-decoded using motion energy features extracted from the action videos and animations (*Appendix 1—figure 1*). This analysis revealed significant above chance decoding accuracy (30%), suggesting that actions and animations indeed contain coincidental visual similarities. To test whether these similarities can explain the effects observed in V1, we used the motion energy decoding matrix as a model for a representational similarity analysis (RSA; see Results section 'Representational geometry of action-structure- and body-motion-related representations').

## Representation of implied versus visible action effects

Humans can recognize many goal-directed actions from mere body movements, as in pantomime. This demonstrates that the brain is capable of inferring the effect that an action has on objects based on the analysis of movement kinematics without the analysis of a visible interaction with an object. Inferring action effects from body movements is easier via pantomimes than with PLD stick figures, because the former provide richer and more fine-grained body information, and in the case of object manipulations, object information (e.g. shape) implied by the pantomimed grasp. Hence, neither pantomimes nor PLDs contain visible information about objects and action effects, but this information is more easily accessible in pantomimes than in PLDs. This difference between pantomimes and PLDs allows testing whether there are brain regions that represent effect structures in the absence of visual information about objects and action effects. We focused on brain regions that revealed the most robust decoding of action effect structures, that is, aIPL and LOTC (see *Figure 4—figure supplement 1* for results in SPL). We first tested for sensitivity to implied effect structures by comparing the cross-decoding of actions and pantomimes (strongly implied hand-object interaction)

with the cross-decoding of actions and PLDs (less implied hand-object interaction). This was the case in both aIPL and LOTC: action-pantomime cross-decoding revealed higher decoding accuracies than action-PLD cross-decoding (*Figure 4A*; all $t$(23) > 3.54, all p < 0.0009; one-tailed). The same pattern should be observed in the comparison of action-pantomime and pantomime-PLD cross-decoding, which was indeed the case (*Figure 3A*; all $t$(23) > 2.96, all p < 0.0035; one-tailed). These findings suggest that the representation of action effect structures in aIPL does not require a visible inter-action with an object. However, the higher decoding across actions and pantomimes might also be explained by the higher visual and kinematic similarity between actions and pantomimes, in partic-ular the shared information about hand posture and hand movements, which are not present in the PLDs. A more selective test is therefore the comparison of animation-pantomime and animation-PLD cross-decoding: as the animations do not provide any body-related information, a difference can only be explained by the stronger matching of effect structures between animations and pantomimes. We found higher cross-decoding for animation-pantomime versus animation-PLD in right aIPL and bilateral LOTC (all $t$(23) > 3.09, all p < 0.0025; one-tailed), but not in left aIPL ($t$(23) = 0.73, p = 0.23, one-tailed). However, a repeated measures ANOVA revealed no significant interaction between TEST (animation-pantomime, animation-PLD) and ROI (left aIPL, right aIPL; $F$(1,23) = 3.66, p = 0.068), precluding strong conclusions about differential sensitivity of right vs. left aIPL in representing implied action effects.

Together, these results suggest that right aIPL and bilateral LOTC are sensitive to implied action effects. This finding was also obtained in a whole-brain conjunction analysis, which revealed effects in right aIPL and bilateral LOTC but not in other brain regions (*Figure 4B*).

## Representational content of brain regions sensitive to action effect structures and body motion

To explore in more detail what types of information were isolated by the action-animation and action-PLD cross-decoding, we performed an RSA.

We first focus on the representations identified by the action-animation decoding. To charac-terize the representational content in the ROIs, we extracted the classification matrices of the action-animation decoding from the ROIs (*Figure 5A*) and compared them with different similarity models (*Figure 5B*) using multiple regression. Specifically, we aimed at testing at which level of granularity action effect structures are represented in aIPL and LOTC: Do these regions encode the broad type of action effects (change of shape, change of location, ingestion) or do they encode specific action effects (compression, division, etc.)? In addition, we aimed at testing whether the effects observed in EVC can be explained by a motion energy model that captures the similarities between actions and animations that we observed in the stimulus-based action-animation decoding using motion energy features. We therefore included V1 in the ROI analysis. We found clear evidence that the representa-tional content in right aIPL and bilateral LOTC can be explained by the effect type model but not by the action-specific model (*Figure 5C*; all two-sided paired $t$-tests between models p < 0.005). In left V1, we found that the motion energy model could indeed explain some representational variance; however, in both left and right V1 we also found effects for the effect type model. We assume that there were additional visual similarities between the broad types of actions and animations that were not captured by the motion energy model (or other visual models; see *Appendix 1—figure 1*). A searchlight RSA revealed converging results, and additionally found effects for the effect type model in the ventral part of left aIPL and for the action-specific model in the left anterior temporal lobe, left dorsal central gyrus, and right EVC (*Figure 5D*). The latter findings were unexpected and should be interpreted with caution, as these regions (except right EVC) were not found in the action-animation cross-decoding and therefore should not be considered reliable (*Ritchie et al., 2017*). The motion energy model did not reveal effects that survived the correction for multiple comparison, but a more lenient uncorrected threshold of p = 0.005 revealed clusters in left EVC and bilateral posterior SPL.

To characterize the representations identified by the action-PLD cross-decoding, we used a model of manuality that captures whether the actions are unimanual or bimanual, an action-specific model as used in the action-animation RSA above, and a kinematics model that was based on the three-dimensional (3D) kinematic marker positions of the PLDs (*Figure 6B*). Since pSTS is a key region for biological motion perception, we included this region in the ROI analysis. The manuality model explained the representational variance in the parietal ROIs, pSTS, and LOTC, but not in V1

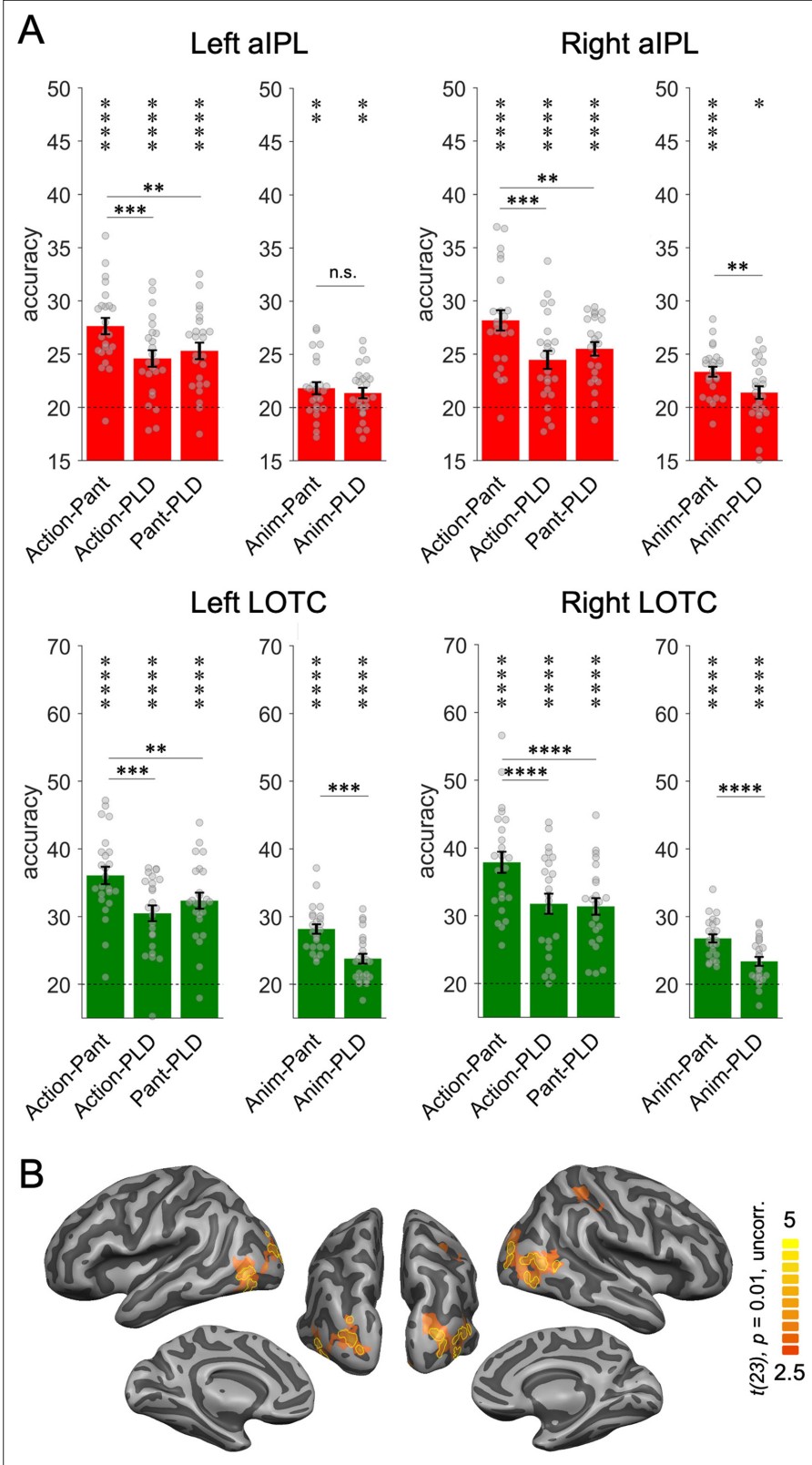

**Figure 4.** Cross-decoding of implied action effect structures. (**A**) Region of interest (ROI) analysis. Cross-decoding schemes involving pantomimes but not point-light-displays (PLDs) (*action-pantomime*, *animation-pantomime*) reveal stronger effects in right anterior inferior parietal lobe (aIPL) than cross-decoding schemes involving PLDs (*action-PLD*, *pantomime-PLD*, *animation-PLD*), suggesting that action effect structure representations in right

*Figure 4 continued on next page*

*Figure 4 continued*

aIPL respond to implied object manipulations in pantomime irrespective of visuospatial processing of observable object state changes. Same conventions as in *Figure 3*. (**B**) Conjunction of the contrasts *action-pantomime versus action-PLD*, *action-pantomime versus pantomime-PLD*, and *animation-pantomime versus animation-PLD*. Uncorrected t-map thresholded at p = 0.01; yellow outlines indicate clusters surviving Monte-Carlo correction for multiple comparisons (voxel threshold p = 0.001, corrected cluster threshold p = 0.05).

The online version of this article includes the following figure supplement(s) for figure 4:

**Figure supplement 1.** Cross-decoding of implied action effect structures in superior parietal lobe (SPL).

(*Figure 6C*; all two-sided paired *t*-tests between V1 and other ROIs p < 0.002). By contrast, the action-specific model revealed significant effects in V1 and LOTC, but not in pSTS and parietal ROIs (but note that effects in V1 and pSTS did not differ significantly from each other; all other two-sided paired *t*-tests between mentioned ROIs were significant at p < 0.0005). The kinematics model explained the representational variance in all ROIs. A searchlight RSA revealed converging results, and additionally found effects for the manuality model in bilateral dorsal/medial prefrontal cortex and in right ventral prefrontal cortex and insula (*Figure 6D*).

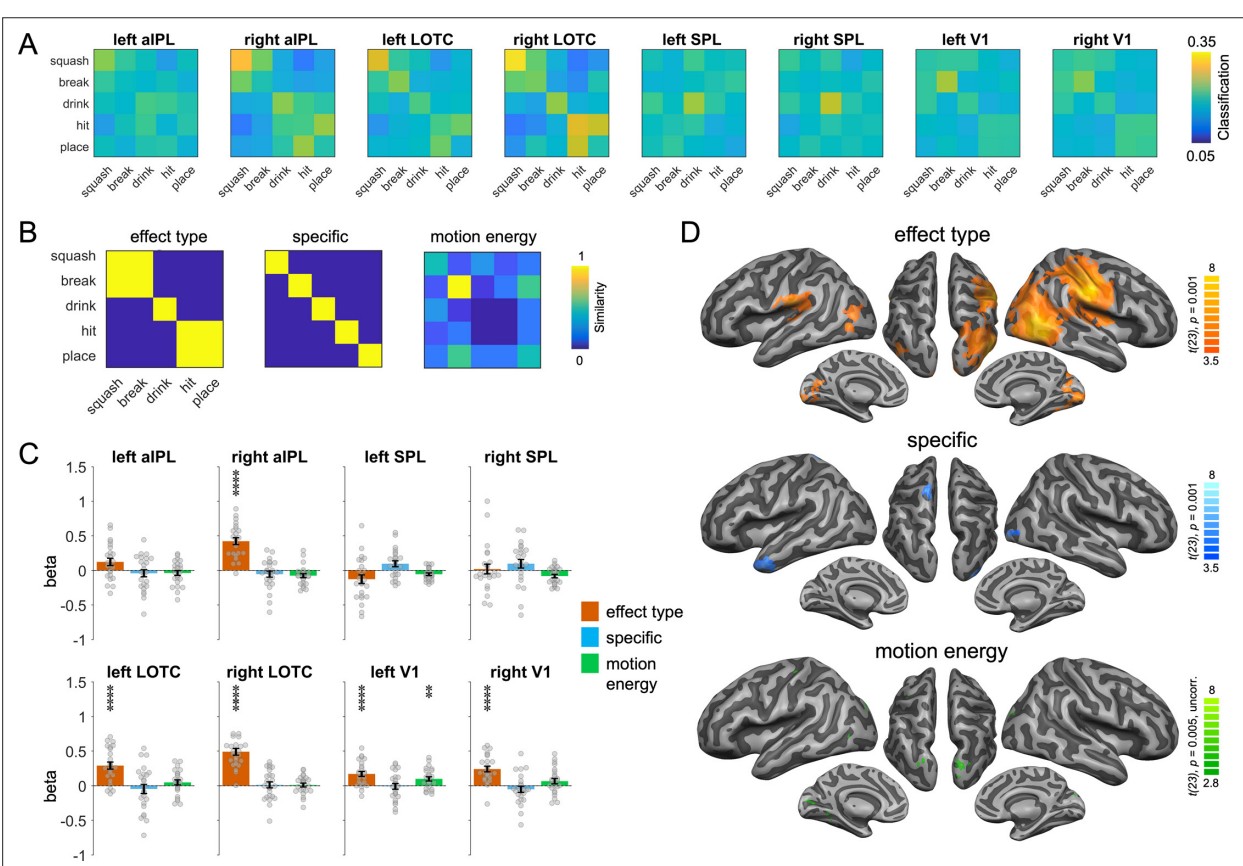

**Figure 5.** Representational similarity analysis (RSA) for the action-animation representations. (**A**) Classification matrices of regions of interest (ROIs). (**B**) Similarity models used in the RSA. (**C**) Multiple regression RSA ROI analysis. Asterisks indicate FDR-corrected significant decoding accuracies above chance (**p < 0.01, ****p < 0.0001). Error bars indicate SEM (N=24). (**D**) Multiple regression RSA searchlight analysis. T-maps are thresholded using Monte-Carlo correction for multiple comparisons (voxel threshold p = 0.001, corrected cluster threshold p = 0.05) except for the motion energy model (p = 0.005, uncorrected).

The online version of this article includes the following figure supplement(s) for figure 5:

**Figure supplement 1.** Cross-decoding results for Action-Pantomime, Pantomime-PLD, Animation-Pantomime, and Animation-PLD.

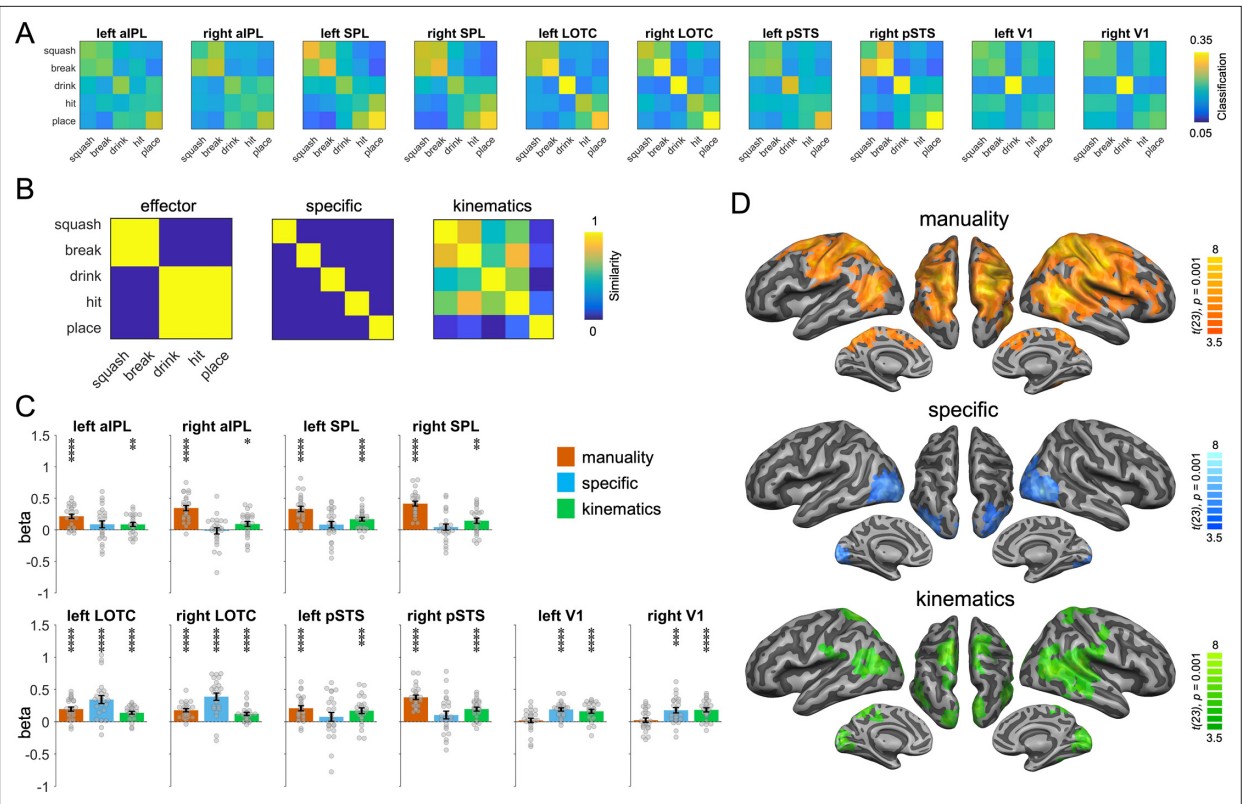

**Figure 6.** Representational similarity analysis (RSA) for the action-PLD representations. (**A**) Classification matrices of regions of interest (ROIs). (**B**) Similarity models used in the RSA. (**C**) Multiple regression RSA ROI analysis. (**D**) Multiple regression RSA searchlight analysis. Same figure conventions as in *Figure 5*.

## Representational similarity between brain regions

Finally, we investigated how similar the ROIs were with regard to the representational structure obtained by the action-animation and action-PLD cross-decoding. To this end, we correlated the classification matrices for both decoding schemes and all ROIs with each other (*Figure 7A*) and displayed the similarities between them using multidimensional scaling (*Figure 7B*) and dendrograms (*Figure 7C*).

The aim for this analysis was twofold: First, we wanted to assess for the action-animation decoding how the representational structure in V1 relates to LOTC and aIPL. If V1 is representationally similar to LOTC and aIPL, this might point toward potential visual factors that drove the cross-decoding in visual

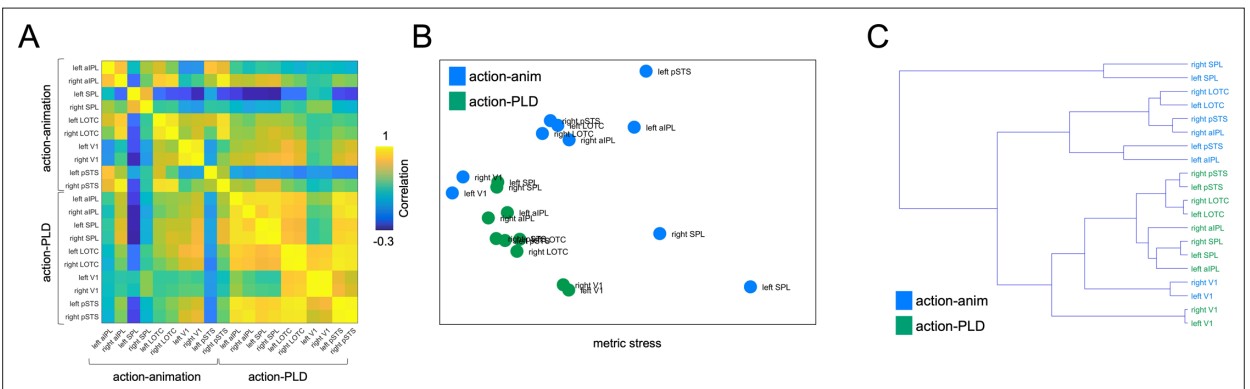

**Figure 7.** Region of interest (ROI) similarity for action-animation and action-PLD representations. (**A**) Correlation matrix for the action-animation and action-PLD decoding and all ROIs. (**B**) Multidimensional scaling. (**C**) Dendrogram plot.

cortex but potentially also in higher-level LOTC and aIPL. However, this was not the case: The V1 ROIs formed a separate cluster that was distinct from a cluster formed by aIPL and LOTC. This suggests that the V1 represents different information than aIPL and LOTC.

Second, we aimed at testing whether the effects in aIPL for the action-PLD decoding reflect the representation of action effect structures or rather representations related to body motion. In the former case, the representational organization in aIPL should be similar for the action-animation and action-PLD cross-decoding. In the latter case, the representational organization for action-PLD should be similar between aIPL and the other ROIs. We found that for the action-PLD decoding, all ROIs were clustered relatively closely together, and aIPL did not show similarity to the action-animation ROIs, specifically to aIPL. This finding argues against the interpretation that the effects in aIPL for the action-PLD cross-decoding were driven by action effect structures. Rather, it suggests that aIPL also encodes body motion, which is in line with the RSA results reported in the previous section.

## Discussion

We provide evidence for neural representations of action effect structures in aIPL and LOTC that generalize between perceptually highly distinct stimulus types – naturalistic actions and abstract animations. The representation of effect structures in aIPL is distinct from parietal representations of body movements, which were predominantly located in SPL. While body movement representations are generally bilateral, action effect structure representations are lateralized to the right aIPL and LOTC. In right aIPL and bilateral LOTC, action effect structure representations do not require a visible interaction with objects but also respond to action effects implied by pantomime.

Recognizing goal-directed actions requires a processing stage that captures the effect an action has on a target entity. Using cross-decoding between actions and animations, we found that aIPL and LOTC encode representations that are sensitive to the core action effect structure, that is, the type of change induced by the action. As the animations did not contain biological motion or specific object information matching the information in the action videos, these representations are independent of specific motion characteristics and object identity. This suggests an abstract level of representation of visuospatial and temporal relations between entities and their parts that may support the identification of object change independently of specific objects (e.g. dividing, compressing, ingesting, or moving something). Object-generality is an important feature as it enables the recognition of action effects on novel, unfamiliar objects. This type of representation fits the idea of a more general neural mechanism supporting mechanical reasoning about how entities interact with, and have effects on, each other (*Fischer et al., 2016*; *Karakose-Akbiyik et al., 2023*). Using multiple regression RSA, we showed that action effect structure representations in these regions capture the broad action effect type, that is, a change of shape/configuration, a change of location, and ingestion. However, since this analysis was based on only five actions, a more comprehensive investigation is needed to understand the organization of a broader range of action effect types (see also *Worgotter et al., 2013*).

In right aIPL and bilateral LOTC, the representation of action effect structures did not depend on a visible interaction with objects but could also be activated by pantomime, that is, an implied interaction with objects. This suggests that right aIPL and LOTC do not merely represent temporospatial relations of entities in a perceived scene. Rather, the effects in these regions might reflect a more inferential mechanism critical for understanding hypothetical effects of an interaction on a target object.

Interestingly, action effect structures appear lateralized to the right hemisphere. This is in line with the finding that perception of cause–effect relations, for example, estimating the effects of colliding balls, activates right aIPL (*Fugelsang et al., 2005*; *Straube and Chatterjee, 2010*). However, in the context of action recognition, the involvement of aIPL is usually bilateral or sometimes left-lateralized, in particular for actions involving an interaction with objects (*Caspers et al., 2010*). Also, mechanical reasoning about tools – the ability to infer the effects of tools on target objects based on the physical properties of tools and objects, such as shape and weight – is usually associated with left rather than right aIPL (*Goldenberg and Spatt, 2009*; *Reynaud et al., 2016*; *Leshinskaya et al., 2020*). Thus, left and right aIPL appear to be disproportionally sensitive to different structural aspects of actions and events: Left aIPL appears to be more sensitive to the type of interaction between entities, that is, how a body part or an object exerts a force onto a target object (e.g. how a hand makes contact with an object to push it), whereas right aIPL appears to be more sensitive to the effect induced by that interaction (the displacement of the object following the push). In our study, the animations contained

interactions, but they did not show precisely how a force was exerted onto the target object that led to the specific effects: In all animations, the causer made contact with the target object in the same manner. Thus, the interaction could not drive the cross-decoding between actions and animations. Only the effects – the object changes – differed and could therefore be discriminated by the classification. Two questions arise from this interpretation: Would similar effects be observed in right aIPL (and LOTC) if the causer were removed, so that only the object change were shown in the animation? And would effects be observed in the left aIPL for distinguishable interactions (e.g. a triangle hitting a target object with the sharp or the flat side), perhaps even in the absence of the induced effect (dividing or compressing object, respectively)?

Action effect representations were found not only in aIPL but also LOTC. Interestingly, the RSA did not reveal substantially different representational content – both regions are equally sensitive to the effect type and their representational organization in response to the five action effects used in this experiment is highly similar. As it appears unlikely that aIPL and LOTC represent identical information, this raises the question of what different functions these regions provide in the context of action effect representation. Right LOTC is associated with the representation of socially relevant information, such as faces, body parts, and their movements (*Chao et al., 1999*; *Pitcher and Ungerleider, 2021*). Our findings suggest that right LOTC is not only sensitive to the perception of body-related information but also to body-independent information important for action recognition, such as object change. It remains to be investigated whether there is a dissociation between the action-independent representation of mere object change (e.g. in shape or location) and a higher-level representation of object change as an effect of an action. Left LOTC is sensitive to tools and effectors (*Bracci and Peelen, 2013*), which might point toward a role in representing putative causes of the observed object changes. Moreover, action representations in left LOTC are perceptually more invariant, as they can be activated by action verbs (*Watson et al., 2013*), generalize across vision and language (*Wurm and Caramazza, 2019b*), and more generally show signatures of conceptual representation (*Lingnau and Downing, 2015*; *Wurm and Caramazza, 2022*). Thus, left LOTC might have generalized across actions and animations at a conceptual, possibly propositional level (e.g. the meaning of dividing, compressing, etc.), rather than at a structural level. Notably, conceptual action representations are typically associated with left anterior LOTC, but not right LOTC and aIPL, which argues against the interpretation that the action-animation cross-decoding in right LOTC and aIPL captured conceptual action representations rather than structural representations of the temporospatial object change type. From a more general perspective, cross-decoding between different stimulus types and formats might be a promising approach to address the fundamental question of whether the format of certain representations is propositional (*Pylyshyn, 2003*) or depictive (*Kosslyn et al., 2006*; *Martin, 2016*).

In contrast to the perceptually general representation of action effect structures in aIPL, the representation of body movements is more specific in terms of visuospatial relations between involved elements, that is, body parts. These representations were predominantly found in bilateral SPL, rather than aIPL, as well as in LOTC. This is in line with previous studies demonstrating stronger decoding of PLD actions in SPL than in aIPL (*Yargholi et al., 2023*) and stronger decoding of human actions as opposed to object events in SPL (*Karakose-Akbiyik et al., 2023*). The RSA revealed that SPL, as well as adjacent regions in parietal cortex including aIPL and LOTC, are particularly sensitive to manuality of the action (uni- vs. bimanual) and the movement kinematics specific to an action. An interesting question for future research is whether movement representation in SPL is particularly tuned to biological motion or equally to similarly complex nonbiological movements. LOTC was not only sensitive to manuality and kinematics, but particularly in discriminating the five actions from each other, which suggests that LOTC is most sensitive to capture even subtle movement differences.

The action-PLD cross-decoding revealed widespread effects in LOTC and parietal cortex, including aIPL. What type of representation drove the decoding in aIPL? One possible interpretation is that aIPL encodes both body movements (isolated by the action-PLD cross-decoding) and action effect structures (isolated by the action-animation cross-decoding). Alternatively, aIPL selectively encodes action effect structures, which have been activated by the PLDs. A behavioral test showed that PLDs at least weakly allow for recognition of the specific actions (*Supplementary file 2*), which might have activated corresponding action effect structure representations. In addition, the finding that aIPL revealed effects for the cross-decoding between animations and PLDs further supports the interpretation that PLDs have activated, at least to some extent, action effect structure representations. On the other

hand, if aIPL encodes *only* action effect structures, we would expect that the representational similarity patterns in aIPL are similar for the action-PLD and action-animation cross-decoding. However, this was not the case; rather, the representational similarity pattern in aIPL was more similar to SPL for the action-PLD decoding, which argues against substantially distinct representational content in aIPL versus SPL for the action-PLD decoding. In addition, the RSA revealed sensitivity to manuality and kinematics also in aIPL, which suggests that the action-PLD decoding in aIPL was at least partially driven by representations related to body movements. Taken together, these findings suggest that aIPL encodes not only action effect structures, but also representations related to body movements. Likewise, also SPL shows some sensitivity to action effect structures, as demonstrated by effects in SPL for the action-animation and pantomime-animation cross-decoding. Thus, our results suggest that aIPL and SPL are not selectively but disproportionally sensitive to action effects and body movements, respectively.

The action effect structure and body movement representations in aIPL and SPL identified here may not only play a role in the recognition of others' actions but also in the execution of goal-directed actions, which requires visuospatial processing of own body movements and of the changes in the world induced by them (*Fischer and Mahon, 2021*). This view is compatible with the proposal that the dorsal 'where/how' stream is subdivided into sub-streams for the visuomotor coordination of body movements in SPL and the manipulation of objects in aIPL (*Rizzolatti and Matelli, 2003*; *Binkofski and Buxbaum, 2013*).

In conclusion, our study dissociated important stages in the visual processing of actions: the representation of body movements and the effects they induce in the world. These stages draw on subregions in parietal cortex – SPL and aIPL – as well as LOTC. These results help to clarify the roles of these regions in action understanding and more generally in understanding the physics of dynamic events. The identification of action effect structure representations in aIPL and LOTC has implications for theories of action understanding: Current theories (see for review e.g. *Zentgraf et al., 2011*; *Kemmerer, 2021*; *Lingnau and Downing, 2024*) largely ignore the fact that the recognition of many goal-directed actions requires a physical analysis of the action-induced effect, that is, a state change of the action target. Moreover, premotor and inferior parietal cortex are usually associated with motor- or body-related processing during action observation. Our results, together with the finding that premotor and inferior parietal cortex are similarly sensitive to actions and inanimate object events (*Karakose-Akbiyik et al., 2023*), suggest that large parts of the 'action observation network' are less specific for body-related processing in action perception than usually thought. Rather, this network might provide a substrate for the physical analysis and predictive simulation of dynamic events in general (*Schubotz, 2007*; *Fischer, 2024*). In addition, our finding that the (body-independent) representation of action effects substantially draws on right LOTC contradicts strong formulations of a 'social perception' pathway in LOTC that is *selectively* tuned to the processing of moving faces and bodies (*Pitcher and Ungerleider, 2021*). The finding of action effect representation in right LOTC/pSTS might also offer a novel interpretation of a right pSTS subregion thought to specialized for social interaction recognition: Right pSTS shows increased activation for the observation of contingent action-reaction pairs (e.g. agent A points toward object; agent B picks up object) as compared to two independent actions (i.e. the action of agent A has no effect on the action of agent B) (*Isik et al., 2017*). Perhaps the activation reflects the representation of a *social* action effect – the change of an agent's state induced by someone else's action. Thus, the representation of action effects might not be limited to physical object changes but might also comprise social effects not induced by a physical interaction between entities. Finally, not all actions induce an observable change in the world. It remains to be tested whether the recognition of, for example, communication (e.g. speaking, gesturing) and perception actions (e.g. observing, smelling) similarly relies on structural action representations in aIPL and LOTC.

## Methods

### Participants

Twenty-four right-handed adults (15 females; mean age, 23.7 years; age range, 20–38 years) participated in this experiment. All participants had normal or corrected-to-normal vision and no history of neurological or psychiatric disease. All procedures were approved by the Ethics Committee for research involving human participants at the University of Trento, Italy (Protocol Nr. 2019-022).

## Stimuli

The stimulus set consisted of videos of five object-directed actions (squashing a plastic bottle, breaking a stick, drinking water, hitting a paper ball, and placing a cup on a saucer) that were shown in four different formats: naturalistic actions, pantomimes, PLD stick figures, and abstract animations (*Figure 2*; informed consent, and consent to publish, was obtained from the actor shown in the figure). The actions were selected among a set of possible actions based on two criteria: (1) The actions should be structurally different from each other as much as possible. (2) The action structures (e.g. of dividing) should be depictable as animations, but at the same time the animations should be associated with the corresponding concrete actions as little as possible to minimize activation of conceptual action representations (e.g. of 'breaking a stick'). The resulting set of five actions belonged to three broad categories of changes: shape/configuration changes (break, squash), location changes (hit, place), and ingestion (drink). This categorization was not planned before designing the study but resulted from the stimulus selection.

For each action and stimulus format, eight exemplars were generated to increase the perceptual variance of the stimuli. All videos were in RGB color, had a length of 2 s (30 frames per second), and a resolution of 400 × 225 pixels.

Naturalistic actions and corresponding pantomimes were performed by two different persons (female, male) sitting on a chair at a table in a neutral setting. The actions were filmed from two different camera viewpoints (approx. 25° and 40°). Finally, each video was mirrored to create left- and right-sided variants of the actions.

For the generation of PLD stick figures, the actions were performed in the same manner as the action videos in a motion-capture lab equipped with a Qualisys motion-capture system (Qualisys AB) comprising 5 ProReflex 1000 infrared cameras (100 frames per second). Thirteen passive kinematic markers (14 mm diameter) were attached to the right and left shoulders, elbows, hands, hips, knees, feet, and forehead of a single actor (the same male actor as in the action and pantomime videos), who performed each action two times. Great care was taken that the actions were performed with the same movements as in the action and pantomime videos. 3D kinematic marker positions were processed using the Qualisys track manager and Biomotion Toolbox V2 (*van Boxtel and Lu, 2013*). Missing marker positions were calculated using the interpolation algorithm of the Qualisys track manager. To increase the recognizability of the body, we connected the points in the PLDs with white lines to create arms, legs, trunk, and neck. PLD stick figures were shown from two angles (25° and 40°), and the resulting videos were left–right mirrored.

Abstract animations were designed to structurally match the five actions in terms of the induced object change. At the same time, they were produced to be as abstract as possible so as to minimize the match at both basic perceptual levels (e.g. shape, motion) and conceptual levels. To clarify the latter, the abstract animation matching the 'breaking' action was designed to be structurally similar (causing an object to divide in half) without activating a specific action meaning such as 'breaking a stick'. In all animations, the agent object (a circle with a smiley) moved toward a target object (a rectangle or a circle). The contact with the target object at 1 s after video onset induced different kinds of effects, that is, the target object broke in half or was compressed, ingested (decreased in size until it disappeared), propelled, or pushed to the side. The animations were created in MATLAB (Mathworks; RRID:SCR_001622) with Psychtoolbox-3 (*Brainard, 1997*). The speeds of all agent and target-object movements were constant across the video. To increase stimulus variance, 8 exemplars per action were generated using two target-object shapes (rectangle and circle), two color schemes for the agent–target pairs (green-blue and pink-yellow), and two action directions (left-to-right and right-to-left).

## Behavioral experiment

To assess how much the animations, PLD stick figures, and pantomimes were associated with the specific action meanings of the naturalistic actions, we performed a behavioral experiment. Fourteen participants observed videos of the animations, PLDs (without stick figures), and pantomimes in three separate sessions (in that order) and were asked to describe what kind of actions the videos depict and to rate their confidence on a Likert scale from 1 (not confident at all) to 10 (very confident). Because the results for PLDs were unsatisfying (several participants did not recognize human motion in the PLDs), we added stick figures to the PLDs as described above and repeated the rating for PLD stick figures with seven new participants, as reported below.

A general observation was that almost no participant used verb–noun phrases (e.g. 'breaking a stick') in their descriptions for all stimulus types. For the animations, the participants used more abstract verbs or nouns to describe the actions (e.g. dividing, splitting, division; *Supplementary file 1*). These abstract descriptions matched the intended action structures quite well, and participants were relatively confident about their responses (mean confidences between 6 and 7.8). These results suggest that the animations were not substantially associated with specific action meanings (e.g. 'breaking a stick') but captured the coarse action structures. For the PLD stick figures (*Supplementary file 2*), responses were more variable and actions were often confused with kinematically similar but conceptually different actions (e.g. breaking --> shaking, hitting --> turning page, squashing --> knitting). Confidence ratings were relatively low (mean confidences between 3 and 5.1). These results suggest that PLD stick figures, too, were not substantially associated with specific action meanings and additionally did not clearly reveal the underlying action effect structures. Finally, pantomimes were recognized much better, which was also reflected in high confidence ratings (mean confidences between 8 and 9.2; *Supplementary file 3*). This suggests that, unlike PLD stick figures, pantomimes allowed much better to access the underlying action effect structures.

## Experimental design

For all four sessions, stimuli were presented in a mixed event-related design. In each trial, videos were followed by a 1-s fixation period. Each of the five conditions was presented four times in a block, intermixed with three catch trials (23 trials per block). Four blocks were presented per run, separated by 8-s fixation periods. Each run started with a 2-s fixation period and ended with a 16-s fixation period. In each run, the order of conditions was pseudorandomized to ensure that each condition followed and preceded each other condition a similar number of times in each run. Each participant was scanned in four sessions (animations, PLDs, pantomimes, actions), each consisting of three functional scans. The order of sessions was chosen to minimize the possibility that participants would associate specific actions/objects with the conditions in the animation and PLD sessions. In other words, during the first session (animations), participants were unaware that they would see human actions in the following sessions; during the second session (PLDs), they were ignorant of the specific objects and hand postures/movements. Each of the five conditions was shown 48 times (4 trials per block × 4 blocks × 3 runs) in each session. Each exemplar of every video was presented six times in the experiment.

## Task

We used a catch-trial-detection task to ensure that participants paid constant attention during the experiment and were not biased to different types of information in the various sessions. Participants were instructed to attentively watch the videos and to press a button with the right index finger on a response-button box whenever a video contained a glitch, that is, when the video did not play smoothly but jerked for a short moment (300 ms). Glitches were created by selecting a random time window of eight video frames of the video (excluding the first 10 and last 4 frames) and shuffling the order of the frames within that window. The task was the same for all sessions. Before fMRI, participants were instructed and trained for the first session only (animations). In all four sessions, the catch trials were identified with robust accuracy (animations: 0.73 ± 0.02 SEM, PLDs: 0.65 ± 0.02, pantomimes: 0.69 ± 0.02, actions: 0.68 ± 0.02). Participants were not informed about the purpose and design of the study before the experiment.

## Data acquisition

Functional and structural data were collected using a 3T Siemens Prisma MRI scanner and a 64-channel head coil. Functional images were acquired with a T2*-weighted gradient echo-planar imaging (EPI) sequence. Acquisition parameters were a repetition time (TR) of 1.5 s, an echo time of 28 ms, a flip angle of 70°, field of view of 200 mm matrix size of 66 × 66, voxel resolution 3 × 3 × 3 mm. We acquired 45 slices in ascending interleaved odd–even order. Each slice was 3 mm thick. There were 211 volumes acquired in each functional run.

Structural T1-weighted images were acquired using an MPRAGE sequence (Slice number = 176, TR = 2.53 s, inversion time = 1.1 s, flip angle = 7°, 256 × 256 mm field of view, 1 × 1 × 1 mm resolution).

## Preprocessing

Data were analyzed using BrainVoyager QX 2.84 (BrainInnovation) in combination with the SPM12 and NeuroElf (BVQXTools) toolboxes and custom software written in Matlab (MathWorks; RRID:SCR_001622). Anatomical scans of individual subjects were normalized to the standard SPM12 EPI template (Montreal Neurological Institute [MNI] stereotactic space). Slice time correction was performed on the functional data followed by a 3D motion correction (trilinear interpolation, with the first volume of the first run of each participant as reference). Functional data were co-registered with the normalized anatomical scans followed by spatial smoothing with a Gaussian kernel of 8 mm full width at half maximum (FWHM) for univariate analysis and 3 mm FWHM for multivariate pattern analysis (MVPA).

## Multivariate pattern classification

For each participant, session, and run, a general linear model was computed using design matrices containing 10 action predictors (2 for each action; based on 8 trials from the first two blocks of a run and the second half from the last two blocks of a run, to increase the number of beta samples for classification), a catch-trial predictor, 6 predictors for each parameter of motion correction (3D translation and rotation), and 6 temporal-drift predictors. Each trial was modeled as an epoch lasting from video onset to offset (2 s). The resulting reference time courses were used to fit the signal time courses of each voxel. Predictors were convolved with a dual-gamma hemodynamic impulse response function. Since there were 3 runs per session, there were thus 6 beta maps per action condition.

Searchlight classification was done on each subject in volume space using a searchlight sphere of 12 mm and an LDA (linear discriminant analysis) classifier, as implemented in the CosmoMVPA toolbox (*Oosterhof et al., 2016*). We also tested for robustness of effects across MVPA parameter choices by running the analysis with different ROI sizes (9 and 15 mm) and a support vector machine (SVM) classifier, which revealed similar findings, that is, all critical findings were also found with the alternative MVPA parameters.

For the within-session analyses, all five action conditions of a given session were entered into a five-way multiclass classification using leave-one-out cross validation, that is, the classifier was trained with five out of six beta patterns per action and was tested with the held-out beta pattern. This was done until each beta pattern was tested. The resulting accuracies were averaged across the six iterations and assigned to the center voxel of the sphere. Within-session decoding analyses were performed as sanity checks to ensure that for all stimulus types, the five actions could be reliably decoded (*Appendix 1—figure 2*). For cross-decoding analyses, a classifier was trained to discriminate the voxel activation patterns associated with the five action conditions from one session (e.g. actions) and tested on its ability to discriminate the voxel activation patterns associated with the five action conditions of another session (e.g. animations). The same was done vice versa (training with animations and testing with actions), and the accuracies were averaged across the two directions (*van den Hurk and Op de Beeck, 2019*) (see *Appendix 1—figure 3* for contrasts between cross-decoding directions). In total, there were six across-session pairs: action-animation, action-PLD, action-pantomime, animation-pantomime, pantomime-PLD, and animation-PLD. For searchlight maps of the latter four decoding schemes see *Figure 5—figure supplement 1*.

For all decoding schemes, a one-tailed, one-sample $t$-test was performed on the resulting accuracy maps to determine which voxels had a decoding accuracy that was significantly above the chance level (20%). The resulting t-maps were corrected for multiple comparisons with Monte-Carlo correction as implemented in CosmoMVPA (*Oosterhof et al., 2016*), using an initial threshold of p = 0.001 at the voxel level, 10,000 Monte-Carlo simulations, and a one-tailed corrected cluster threshold of p = 0.05 ($z$ = 1.65).

Conjunction maps were computed by selecting the minimal $z$-value (for corrected maps) or $t$-value (for uncorrected maps) for each voxel of the input maps.

## ROI analysis

ROIs were based on MNI coordinates of the center locations of Brodmann areas (BA) associated with the aIPL/supramarginal gyrus (BA 40; left: −53, −32, 33; right: 51, −33, 34), SPL (BA 7; left: −18, −61, 55; right: 23, −60, 61), LOTC (BA 19; left: −45, −75, 11; right: 44, −75, 5), and EVC (BA 17; left: −11, −81, 7; right: 11, −78, 9), using the MNI2TAL application of the BioImage Suite WebApp (https://

bioimagesuiteweb.github.io/webapp/). Coordinates for left and right pSTS were taken from a meta analysis for human versus non-human body movements (*Grosbras et al., 2012*). To keep the ROI size constant across the different ROIs, we used spherical ROIs as in our previous studies (*Wurm and Caramazza, 2019b*; *Wurm and Caramazza, 2019a*; *Karakose-Akbiyik et al., 2023*). A visualization of the ROIs projected on the cortical surface can be found in *Appendix 1—figure 4*. For each participant, ROI, and decoding scheme, decoding accuracies from the searchlight analysis were extracted from all voxels within a sphere of 12 mm around the ROI center, averaged across voxels, and entered into one-tailed, one-sample *t*-tests against chance (20%). In addition, paired *t*-tests and repeated measures ANOVAs were conducted to test for the differences between ROIs and different decoding schemes. The statistical results of *t*-tests were FDR-corrected for the number of tests and ROIs (*Benjamini and Yekutieli, 2001*).

## Representational similarity analysis

To analyze the representational content isolated by the action-animation and action-PLD cross-decoding, we performed a multiple regression RSA. First, we extracted classification matrices from the the action-animation and action-PLD cross-decoding maps: For each subject, voxel, and cross-decoding scheme, we extracted the classification matrices, symmetrized them, and rescaled them into values between 0 and 1. For each ROI, we averaged the matrices across voxels. The classification matrices were converted into neural representational dissimilarity matrices (RDMs) by subtracting 1 from them.

The neural RDMs were then compared with model RDMs using a multiple regression RSA. The following models were used for the action-animation RSA: (1) An 'effect type' model that captures the similarity in terms of action effect type (shape/configuration changes: break, squash; location changes: hit, place; and ingestion: drink). (2) A 'specific' model that discriminates each specific action from each other with equal distance. (3) A 'motion energy' model that was based on the stimulus-based action-animation decoding (see *Appendix 1—figure 1*) and that captures the similarity of animations and actions in terms of motion energy. No critical collinearity was observed (variance inflation factors <2.6, condition indices <4, variance decomposition proportions <0.9). The following models were used for the action-PLD RSA: (1) A manuality model that captures whether the actions were carried out with one versus both hands (bimanual: break, squash, drink; unimanual: hit, place). (2) A 'specific' model that discriminates each specific action from each other with equal distance. (3) A kinematic model that was based on marker positions of the PLDs. The kinematics model was constructed by averaging the kinematic data across the 2 exemplars per PLD, vectorizing the 3D marker positions of all time points of the PLDs (3 dimensions × 13 markers × 200 time points), computing the pairwise correlations between the five vectors, and converting the correlations into dissimilarity values by subtracting $1 - r$. No critical collinearity was observed (variance inflation factors <3.2, condition indices <4, variance decomposition proportions <0.96). The multiple regression RSA was first done at the whole-brain level using a searchlight approach: For each subject, voxel, and cross-decoding scheme, we extracted the classification matrices and converted them into neural RDMs as described above. Each neural RDM was entered as dependent variable into a multiple regression, together with the model RDMs as independent variables. Note that we included the on-diagonal values of the neural and model RDMs as they contain interpretable zero points (*Walther et al., 2016*), which is necessary for testing the action-specific model. Resulting beta values were Fisher-transformed and entered into one-tailed one-sample *t*-tests. The resulting *t*-maps were corrected for multiple comparisons with Monte-Carlo correction as described above. For ROI analyses, beta values of the RSA were extracted from the ROIs, averaged across voxels, and entered into statistical tests as described above (see ROI analysis').

## Representational similarity between brain regions

To test how similar the representational content of the ROIs for the action-animation and action-PLD decoding are to each other, we used informational connectivity analysis (*Coutanche and Thompson-Schill, 2013*): First, we correlated the neural RDMs of all ROIs and decoding schemes with each other. Specifically, we included the lower triangle of the RDMs (off-diagonal pairwise distances between actions) and also the on-diagonal values of the RDMs, which contain the correct classifications (*Walther et al., 2016*). This was done to increase the number of informative data in the correlations. We then converted the resulting correlations between ROIs/decoding schemes into distances by

subtracting 1 − $r$, and visualized the pairwise distances between ROIs and decoding schemes using multidimensional scaling (metric stress) and a dendrogram plot following a cluster analysis (nearest distance).

## Acknowledgements

We thank Seoyoung Lee for assistance in preparing the video stimuli, Ingmar de Vries for assistance in preparing the PLD stimuli, Ben Timberlake for proof-reading, and the Caramazza Lab for helpful feedback on the study design and interpretation of results. This research was supported by the Caritro Foundation, Italy.

## Additional information

### Funding

| Funder | Grant reference number | Author |
| --- | --- | --- |
| Caritro Foundation | | Doruk Yiğit Erigüç |

The funders had no role in study design, data collection and interpretation, or the decision to submit the work for publication.

### Author contributions

Moritz F Wurm, Conceptualization, Resources, Data curation, Software, Formal analysis, Supervision, Funding acquisition, Validation, Investigation, Visualization, Methodology, Writing – original draft, Project administration, Writing – review and editing; Doruk Yiğit Erigüç, Data curation, Formal analysis, Investigation, Visualization, Methodology, Writing – original draft, Writing – review and editing

### Author ORCIDs

Moritz F Wurm ⬤ https://orcid.org/0000-0003-4358-9815
Doruk Yiğit Erigüç ⬤ https://orcid.org/0009-0003-8073-8135

### Ethics

Informed consent, and consent to publish, was obtained from all participants. All procedures were approved by the Ethics Committee for research involving human participants at the University of Trento, Italy (Protocol Number 2019-022).

Reviewer #1 (Public review): https://doi.org/10.7554/eLife.98521.3.sa1
Author response https://doi.org/10.7554/eLife.98521.3.sa2

## Additional files

### Supplementary files

Supplementary file 1. Results of behavioral pilot experiment for abstract animations. Verbal descriptions of each participant and mean confidence ratings (from 1 = not at all to 10 = very much ± standard deviations).

Supplementary file 2. Results of behavioral pilot experiment for PLD stick figures. Verbal descriptions of each participant and mean confidence ratings (from 1 = not at all to 10 = very much ± standard deviations).

Supplementary file 3. Results of behavioral pilot experiment for pantomimes. Verbal descriptions of each participant and mean confidence ratings (from 1 = not at all to 10 = very much ± standard deviations).

MDAR checklist

### Data availability

Stimuli, MRI data, and code are deposited at the Open Science Framework (https://osf.io/am346/).

The following dataset was generated:

| Author(s) | Year | Dataset title | Dataset URL | Database and Identifier |
|---|---|---|---|---|
| Wurm MF, Erigüç DY | 2024 | Decoding the physics of observed actions in the human brain | https://osf.io/am346/ | Open Science Framework, am346 |

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

## Appendix 1

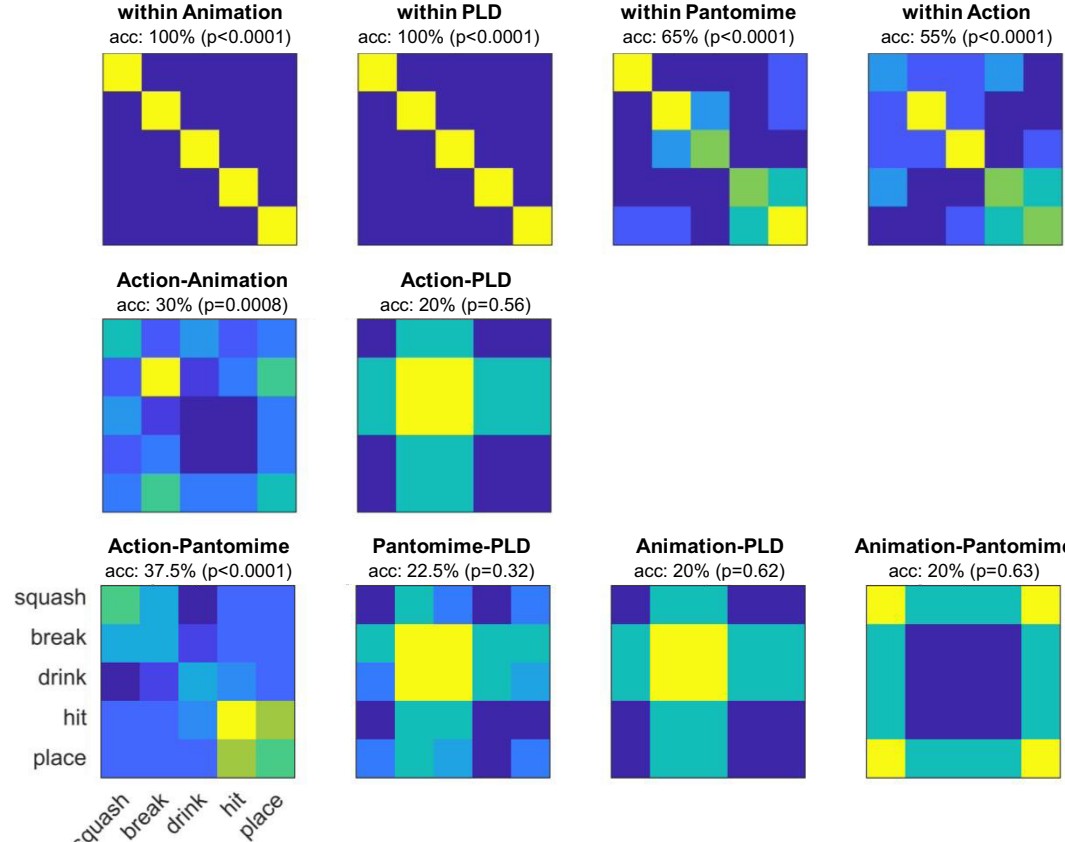

**Appendix 1—figure 1.** Stimulus-based cross-decoding. For each video, motion energy features were extracted as described in ***Nishimoto et al., 2011*** using Matlab code from GitHub (***Nishimoto and Lescroart, 2018***). To reduce the number of features for the subsequent MVPA, we used the motion energy model with 2139 channels. The resulting motion energy features were averaged across time and entered into within- and across-decoding schemes as described for the fMRI-based decoding. For the decoding within stimulus type, we used leave-one-exemplar-out cross-validation, that is, the classifier was train on seven of the eight exemplars for each action and tested on the remaining exemplar, etc. For the cross-decoding, the classifier was trained on all eight exemplars of stimulus type A and tested on all eight exemplars of stimulus type B (and vice versa). Significance was determined using a permutation test: For each decoding scheme, 10,000 decoding tests with shuffled class labels were performed to create a null distribution. p-values were computed by counting the number of values in the null distribution that were greater or as great as the observed decoding accuracy. The within-stimulus-type decoding served as a control analysis and revealed highly significant decoding accuracies for each stimulus type (animations: 100%, PLDs: 100%, pantomimes: 65%, actions: 55%), which suggests that the motion energy data generally contains information that can be detected by a classifier. The cross-decoding between stimulus types was significantly above chance for action-animation and action-pantomime, but not significantly different from chance for the remaining decoding schemes. Interestingly, all cross-decoding schemes with PLDs did not perform well and revealed similar classification matrices (systematically confusing *squash*, *hit*, and *place* with *break* and *drink*). This might be due to different feature complexity and motion information at different spatial frequencies for PLDs, which do not generalize to the other stimulus types. We also tested whether the different stimulus types can be cross-decoded using other visual features. To test for pixelwise similarities, we averaged the video frames of each video, vectorized and z-scored them, and entered them into the decoding schemes as described above. We found above chance decoding for all within-stimulus type schemes, but not for the cross-decoding schemes (animations: 55%, PLDs: 80%, pantomimes: 40%, actions: 30%, action-anim: 15%, action-PLD: 20%, action-pant: 20%, pant-PLD: 12.5%, anim-PLD: 20%, anim-pant: 22.5%). To test for local motion similarities, we computed frame-to-frame optical flow vectors (using Matlab's opticalFlowHS function) for each video, averaged the resulting optical flow values across frames, vectorized and z-scored them, and entered them into the decoding schemes as described above. We found above chance decoding for all within-stimulus type schemes and the action-pantomime cross-decoding, but not for the other cross-decoding schemes (animations: 75%, PLDs: 100%,

pantomimes: 65%, actions: 40%, action-anim: 15%, action-PLD: 18.7%, action-pant: 38.7%, pant-PLD: 26.2%, anim-PLD: 22.5%, anim-pant: 21.2%).

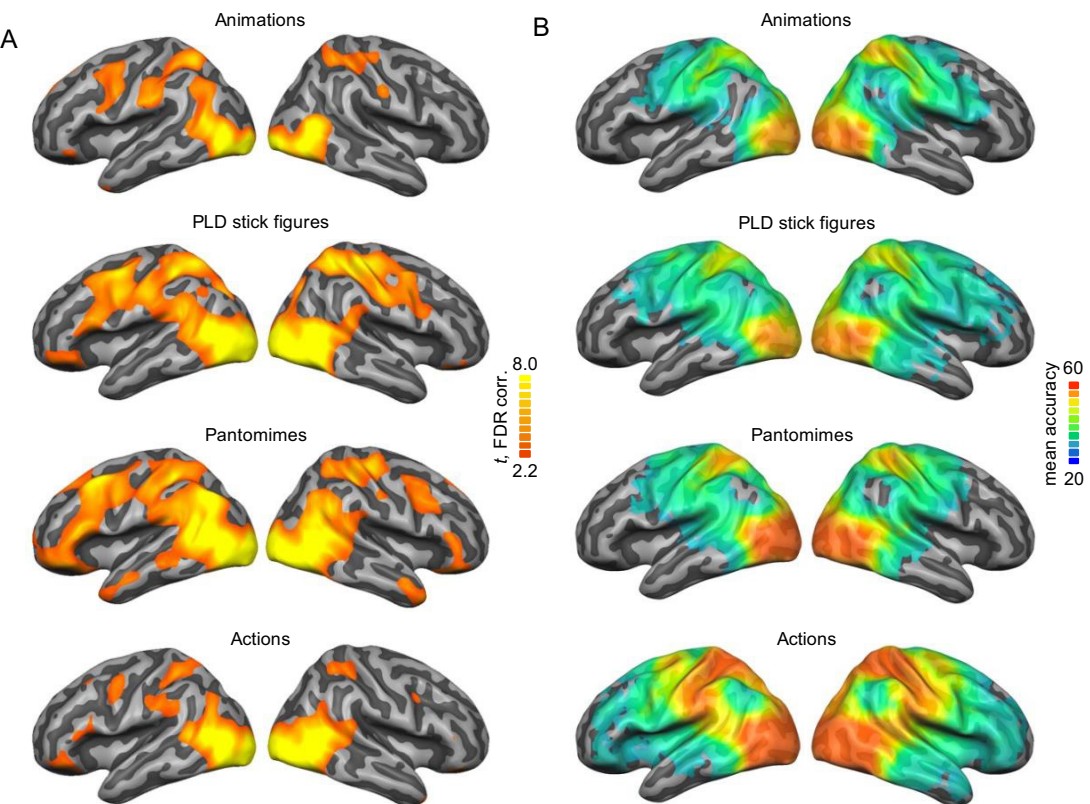

**Appendix 1—figure 2.** Univariate baseline and within-session decoding maps. (**A**) Univariate activation maps for each session (all five actions vs. Baseline; FDR-corrected at p = 0.05) and (**B**) within-session decoding maps (Monte-Carlo corrected for multiple comparisons; voxel threshold p = 0.001, corrected cluster threshold p = 0.05).

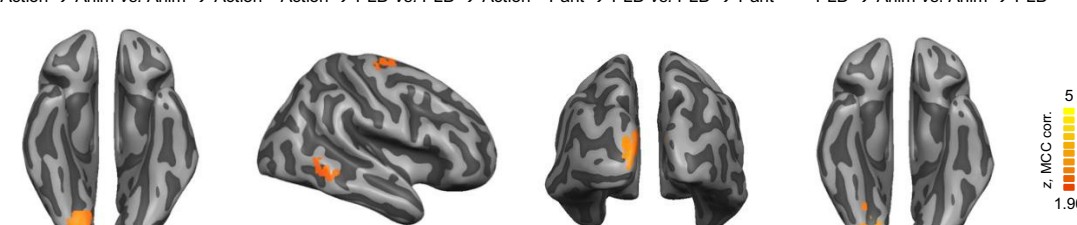

**Appendix 1—figure 3.** Direction-specific cross-decoding effects. To test whether there were differences between the two directions in the cross-decoding analyses, we ran, for each of the six across-session decoding schemes, two-tailed paired samples *t*-tests between the decoding maps of one direction (e.g. action → animation) versus the other direction (animation → action). Direction effects were observed in left early visual cortex for the directions action → animation, PLD → animation, and pantomime → PLD, as well in right middle temporal gyrus and dorsal premotor cortex for action → PLD. These effects might be due to noise differences between stimulus types (***van den Hurk and Op de Beeck, 2019***) and do not affect the interpretation of direction-averaged cross-decoding effects in the main text. Monte-Carlo corrected for multiple comparisons; voxel threshold p = 0.001, corrected cluster threshold p = 0.05.

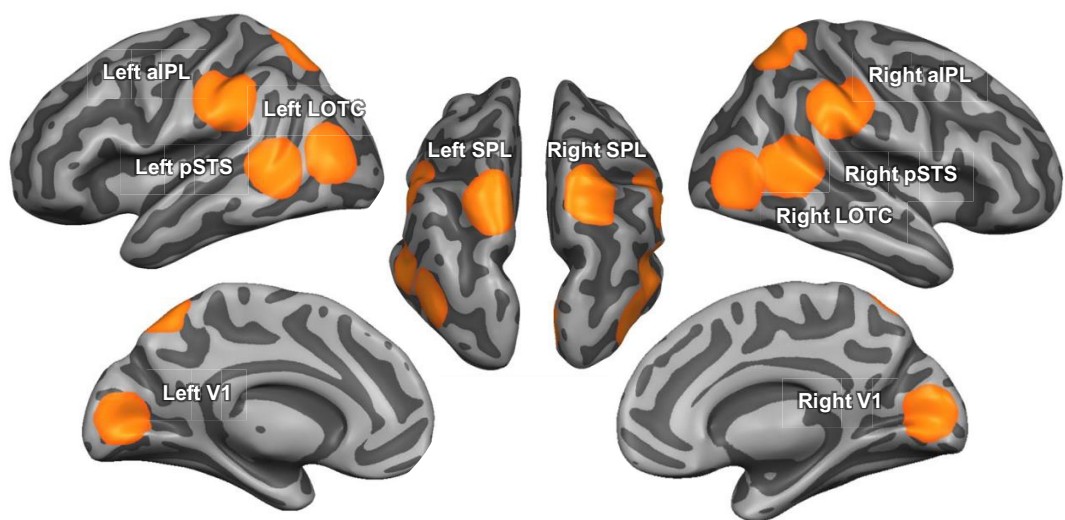

**Appendix 1—figure 4.** ROIs used in the study. Spherical ROIs were in volume space (12 mm radius); here, we projected them on the cortical surface for a better comparison with the whole-brain maps in the main article.

