## [Editor Report · eLife Assessment]

In an **important** fMRI study with an elegant experimental design and rigorous cross-decoding analyses, this work shows a **convincing** dissociation between two parietal regions in visually processing actions. Specifically, aIPL is found to be sensitive to the causal effects of observed actions, while SPL is sensitive to the patterns of body motion involved in those actions. The work will be of broad interest to cognitive neuroscientists, particularly vision and action researchers.

---

## [Referee Report · Reviewer #1 (Public review)]

Summary:

The authors report a study aimed at understanding the brain's representations of viewed actions, with a particular aim to distinguish regions that encode observed body movements, from those that encode the effects of actions on objects. They adopt a cross-decoding multivariate fMRI approach, scanning adult observers who viewed full-cue actions, pantomimes of those actions, minimal skeletal depictions of those actions, and abstract animations that captured analogous effects to those actions. Decoding across different pairs of these action conditions allowed the authors to pull out the contributions of different action features in a given region's representation. The main hypothesis, which was largely confirmed, was that the superior parietal lobe (SPL) more strongly encodes movements of the body, whereas the anterior inferior parietal lobe (aIPL) codes for action effects of outcomes. Specifically, region of interest analyses showed dissociations in the successful cross-decoding of action category across full-cue and skeletal or abstract depictions. Their analyses also highlight the importance of the lateral occipito-temporal cortex (LOTC) in coding action effects. They also find some preliminary evidence about the organisation of action kinds in the regions examined, and take some steps to distinguishing the differences and similarities of action-evoked patterns in primary visual cortex and the other examined regions.

Strengths:

The paper is well-written, and it addresses a topic of emerging interest where social vision and intuitive physics intersect. The use of cross-decoding to examine actions and their effects across four different stimulus formats is a strength of the study. Likewise the a priori identification of regions of interest (supplemented by additional full-brain analyses) is a strength. Finally, the authors successfully deployed a representational-similarity approach that provides more detailed evidence about the different kinds of action features that seem to be captured in each of the regions that were examined.

Weaknesses:

Globally, the findings provide support for the predicted anatomical distinctions, and for the distinction between body-focused representations of actions and more abstract "action effect structures". Viewed more narrowly, the picture is rather complex, and the patterns of (dis)similarity in the activity evoked by different action kinds do not always divide neatly. Probably, examining many more kinds of actions with the multi-format decoding approach developed here will be needed to more effectively disentangle the various contributions of movement, posture, low-level visual properties, and action outcomes/effects.

---

## [Author Response]

The following is the authors’ response to the original reviews.

**eLife Assessment**
In an important fMRI study with an elegant experimental design and rigorous cross-decoding analyses, this work shows a solid dissociation between two parietal regions in visually processing actions. Specifically, aIPL is found to be sensitive to the causal effects of observed actions, while SPL is sensitive to the patterns of body motion involved in those actions. Additional analysis and explanation would help to determine the strength of evidence and the mechanistic underpinnings would benefit from closer consideration. Nevertheless, the work will be of broad interest to cognitive neuroscientists, particularly vision and action researchers.

We thank the editor and the reviewers for their assessment and their excellent comments and suggestions. We really believe they helped us to provide a stronger and more nuanced paper. In our revision, we addressed all points raised by the reviewers. Most importantly, we added a new section on a series of analyses to characterize in more detail the representations isolated by the action-animation and action-PLD cross-decoding. Together, these analyses strengthen the conclusion that aIPL and LOTC represent action effect structures at a categorical rather than specific level, that is, the type of change (e.g., of location or configuration) rather than the specific effect type (e.g. division, compression). SPL is sensitive to body-specific representations, specifically manuality (unimanual vs. bimanual) and movement kinematics. We also added several other analyses and addressed each point of the reviewers. Please find our responses below.

**Public Reviews:**

**Reviewer #1 (Public Review):**
Summary:The authors report a study aimed at understanding the brain's representations of viewed actions, with a particular aim to distinguish regions that encode observed body movements, from those that encode the effects of actions on objects. They adopt a cross-decoding multivariate fMRI approach, scanning adult observers who viewed full-cue actions, pantomimes of those actions, minimal skeletal depictions of those actions, and abstract animations that captured analogous effects to those actions. Decoding across different pairs of these actions allowed the authors to pull out the contributions of different action features in a given region's representation. The main hypothesis, which was largely confirmed, was that the superior parietal lobe (SPL) more strongly encodes movements of the body, whereas the anterior inferior parietal lobe (aIPL) codes for action effects of outcomes. Specifically, region of interest analyses showed dissociations in the successful cross-decoding of action category across full-cue and skeletal or abstract depictions. Their analyses also highlight the importance of the lateral occipito-temporal cortex (LOTC) in coding action effects. They also find some preliminary evidence about the organisation of action kinds in the regions examined.Strengths:The paper is well-written, and it addresses a topic of emerging interest where social vision and intuitive physics intersect. The use of cross-decoding to examine actions and their effects across four different stimulus formats is a strength of the study. Likewise, the a priori identification of regions of interest (supplemented by additional full-brain analyses) is a strength.Weaknesses:I found that the main limitation of the article was in the underpinning theoretical reasoning. The authors appeal to the idea of "action effect structures (AES)", as an abstract representation of the consequences of an action that does not specify (as I understand it) the exact means by which that effect is caused, nor the specific objects involved. This concept has some face validity, but it is not developed very fully in the paper, rather simply asserted. The authors make the claim that "The identification of action effect structure representations in aIPL has implications for theories of action understanding" but it would have been nice to hear more about what those theoretical implications are. More generally, I was not very clear on the direction of the claim here. Is there independent evidence for AES (if so, what is it?) and this study tests the following prediction, that AES should be associated with a specific brain region that does not also code other action properties such as body movements? Or, is the idea that this finding -- that there is a brain region that is sensitive to outcomes more than movements -- is the key new evidence for AES?

Thank you for raising this important issue. We reasoned that AES should exist to support the recognition of perceptually variable actions, including those that we have never experienced before. To the best of our knowledge, there is only indirect evidence for the existence of AES, namely that humans effortlessly and automatically recognize actions (and underlying intentions and feelings) in movements of abstract shapes, as in the famous Heider and Simmel (1949) animations. As these animations do not contain any body posture or movement information at all, the only available cues are the spatiotemporal relations between entities and entity parts in the perceived scene. We think that the effortless and automatic attribution of actions to these stimuli points toward an evolutionary optimized mechanism to capture action effect structures from highly variable action instantiations (so general that it even works for abstract animations). Our study thus aimed to test for the existence of such a level of representation in the brain. We clarified this point in the introduction.

In our revised manuscript, we also revised our discussion of the implications of the finding of AES representations in the brain:

"The identification of action effect structure representations in aIPL and LOTC has implications for theories of action understanding: Current theories (see for review e.g. Zentgraf et al., 2011; Kemmerer, 2021; Lingnau and Downing, 2024) largely ignore the fact that the recognition of many goal-directed actions requires a physical analysis of the action-induced effect, that is, a state change of the action target. Moreover, premotor and inferior parietal cortex are usually associated with motor- or body-related processing during action observation. Our results, together with the finding that premotor and inferior parietal cortex are similarly sensitive to actions and inanimate object events (Karakose-Akbiyik et al., 2023), suggest that large parts of the 'action observation network' are less specific for body-related processing in action perception than usually thought. Rather, this network might provide a substrate for the physical analysis and predictive simulation of dynamic events in general (Schubotz, 2007; Fischer, 2024). In addition, our finding that the (body-independent) representation of action effects substantially draws on right LOTC contradicts strong formulations of a 'social perception' pathway in LOTC that is selectively tuned to the processing of moving faces and bodies (Pitcher and Ungerleider, 2021). The finding of action effect representation in right LOTC/pSTS might also offer a novel interpretation of a right pSTS subregion thought to specialized for social interaction recognition: Right pSTS shows increased activation for the observation of contingent action-reaction pairs (e.g. agent A points toward object; agent B picks up object) as compared to two independent actions (i.e., the action of agent A has no effect on the action of agent B) (Isik et al., 2017). Perhaps the activation reflects the representation of a social action effect - the change of an agent's state induced by someone else's action. Thus, the representation of action effects might not be limited to physical object changes but might also comprise social effects not induced by a physical interaction between entities. Finally, not all actions induce an observable change in the world. It remains to be tested whether the recognition of, e.g., communication (e.g. speaking, gesturing) and perception actions (e.g. observing, smelling) similarly relies on structural action representations in aIPL and LOTC"

On a more specific but still important point, I was not always clear that the significant, but numerically rather small, decoding effects are sufficient to support strong claims about what is encoded or represented in a region. This concern of course applies to many multivariate decoding neuroimaging studies. In this instance, I wondered specifically whether the decoding effects necessarily reflected fully five-way distinction amongst the action kinds, or instead (for example) a significantly different pattern evoked by one action compared to all of the other four (which in turn might be similar). This concern is partly increased by the confusion matrices that are presented in the supplementary materials, which don't necessarily convey a strong classification amongst action kinds. The cluster analyses are interesting and appear to be somewhat regular over the different regions, which helps. However: it is hard to assess these findings statistically, and it may be that similar clusters would be found in early visual areas too.

We agree that in our original manuscript, we did not statistically test what precisely drives the decoding, e.g., specific actions or rather broader categories. In our revised manuscript, we included a representational similarity analysis (RSA) that addressed this point. In short, we found that the action-animation decoding was driven by categorical distinctions between groups of actions (e.g. hit/place vs. the remaining actions) rather than a fully five-way distinction amongst all action kinds. The action-PLD decoding was mostly driven by , specifically manuality (unimanual vs. bimanual) and movement kinematics; in left and right LOTC we found additional evidence for action-specific representations.

Please find below the new paragraph on the RSA:

"To explore in more detail what types of information were isolated by the action-animation and action-PLD cross-decoding, we performed a representational similarity analysis.

We first focus on the representations identified by the action-animation decoding. To inspect and compare the representational organization in the ROIs, we extracted the confusion matrices of the action-animation decoding from the ROIs (Fig. 5A) and compared them with different similarity models (Fig. 5B) using multiple regression. Specifically, we aimed at testing at which level of granularity action effect structures are represented in aIPL and LOTC: Do these regions encode the broad type of action effects (change of shape, change of location, ingestion) or do they encode specific action effects (compression, division, etc.)? In addition, we aimed at testing whether the effects observed in EVC can be explained by a motion energy model that captures the similarities between actions and animations that we observed in the stimulus-based action-animation decoding using motion energy features. We therefore included V1 in the ROI analysis. We found clear evidence that the representational content in right aIPL and bilateral LOTC can be explained by the effect type model but not by the action-specific model (all p < 0.005; two-sided paired t-tests between models; Fig. 5C). In left V1, we found that the motion energy model could indeed explain some representational variance; however, in both left and right V1 we also found effects for the effect type model. We assume that there were additional visual similarities between the broad types of actions and animations that were not captured by the motion energy model (or other visual models; see Supplementary Information). A searchlight RSA revealed converging results, and additionally found effects for the effect type model in the ventral part of left aIPL and for the action-specific model in the left anterior temporal lobe, left dorsal central gyrus, and right EVC (Fig. 5D). The latter findings were unexpected and should be interpreted with caution, as these regions (except right EVC) were not found in the action-animation cross-decoding and therefore should not be considered reliable (Ritchie et al., 2017). The motion energy model did not reveal effects that survived the correction for multiple comparison, but a more lenient uncorrected threshold of p = 0.005 revealed clusters in left EVC and bilateral posterior SPL.

To characterize the representations identified by the action-PLD cross-decoding, we used a manuality model that captures whether the actions were performed with both hands vs. one hand, an action-specific model as used in the action-animation RSA above, and a kinematics model that was based on the 3D kinematic marker positions of the PLDs (Fig. 6B). Since pSTS is a key region for biological motion perception, we included this region in the ROI analysis. The manuality model explained the representational variance in the parietal ROIs, pSTS, and LOTC, but not in V1 (all p < 0.002; two-sided paired t-tests between V1 and other ROIs; Fig. 6C). By contrast, the action-specific model revealed significant effects in V1 and LOTC, but not in pSTS and parietal ROIs (but note that effects in V1 and pSTS did not differ significantly from each other; all other two-sided paired t-tests between mentioned ROIs were significant at p < 0.0005). The kinematics model explained the representational variance in all ROIs. A searchlight RSA revealed converging results, and additionally found effects for the manuality model in bilateral dorsal/medial prefrontal cortex and in right ventral prefrontal cortex and insula (Fig. 6D).”

We also included an ROI covering early visual cortex (V1) in our analysis. While there was significant decoding for action-animation in V1, the representational organization did not substantially match the organization found in aIPL and LOTC: A cluster analysis revealed much higher similarity between LOTC and aIPL than between these regions and V1:

(please note that in this analysis we included the action-PLD RDMs as reference, and to test whether aIPL shows a similar representational organization in action-anim and action-PLD; see below)

Given these results, we think that V1 captured different aspects in the action-animation cross-decoding than aIPL and LOTC. We address this point in more detail in our response to the "Recommendations for The Authors".

**Reviewer #2 (Public Review):**
Summary:This study uses an elegant design, using cross-decoding of multivariate fMRI patterns across different types of stimuli, to convincingly show a functional dissociation between two sub-regions of the parietal cortex, the anterior inferior parietal lobe (aIPL) and superior parietal lobe (SPL) in visually processing actions. Specifically, aIPL is found to be sensitive to the causal effects of observed actions (e.g. whether an action causes an object to compress or to break into two parts), and SPL to the motion patterns of the body in executing those actions.To show this, the authors assess how well linear classifiers trained to distinguish fMRI patterns of response to actions in one stimulus type can generalize to another stimulus type. They choose stimulus types that abstract away specific dimensions of interest. To reveal sensitivity to the causal effects of actions, regardless of low-level details or motion patterns, they use abstract animations that depict a particular kind of object manipulation: e.g. breaking, hitting, or squashing an object. To reveal sensitivity to motion patterns, independently of causal effects on objects, they use point-light displays (PLDs) of figures performing the same actions. Finally, full videos of actors performing actions are used as the stimuli providing the most complete, and naturalistic information. Pantomime videos, with actors mimicking the execution of an action without visible objects, are used as an intermediate condition providing more cues than PLDs but less than real action videos (e.g. the hands are visible, unlike in PLDs, but the object is absent and has to be inferred). By training classifiers on animations, and testing their generalization to full-action videos, the classifiers' sensitivity to the causal effect of actions, independently of visual appearance, can be assessed. By training them on PLDs and testing them on videos, their sensitivity to motion patterns, independent of the causal effect of actions, can be assessed, as PLDs contain no information about an action's effect on objects.These analyses reveal that aIPL can generalize between animations and videos, indicating that it is sensitive to action effects. Conversely, SPL is found to generalize between PLDs and videos, showing that it is more sensitive to motion patterns. A searchlight analysis confirms this pattern of results, particularly showing that action-animation decoding is specific to right aIPL, and revealing an additional cluster in LOTC, which is included in subsequent analyses. Action-PLD decoding is more widespread across the whole action observation network.This study provides a valuable contribution to the understanding of functional specialization in the action observation network. It uses an original and robust experimental design to provide convincing evidence that understanding the causal effects of actions is a meaningful component of visual action processing and that it is specifically localized in aIPL and LOTC.Strengths:The authors cleverly managed to isolate specific aspects of real-world actions (causal effects, motion patterns) in an elegant experimental design, and by testing generalization across different stimulus types rather than within-category decoding performance, they show results that are convincing and readily interpretable. Moreover, they clearly took great care to eliminate potential confounds in their experimental design (for example, by carefully ordering scanning sessions by increasing realism, such that the participants could not associate animation with the corresponding real-world action), and to increase stimulus diversity for different stimulus types. They also carefully examine their own analysis pipeline, and transparently expose it to the reader (for example, by showing asymmetries across decoding directions in Figure S3). Overall, this is an extremely careful and robust paper.Weaknesses:I list several ways in which the paper could be improved below. More than 'weaknesses', these are either ambiguities in the exact claims made, or points that could be strengthened by additional analyses. I don't believe any of the claims or analyses presented in the paper show any strong weaknesses, problematic confounds, or anything that requires revising the claims substantially.(1) Functional specialization claims: throughout the paper, it is not clear what the exact claims of functional specialization are. While, as can be seen in Figure 3A, the difference between action-animation cross-decoding is significantly higher in aIPL, decoding performance is also above chance in right SPL, although this is not a strong effect. More importantly, action-PLD cross-decoding is robustly above chance in both right and left aIPL, implying that this region is sensitive to motion patterns as well as causal effects. I am not questioning that the difference between the two ROIs exists - that is very convincingly shown. But sentences such as "distinct neural systems for the processing of observed body movements in SPL and the effect they induce in aIPL" (lines 111-112, Introduction) and "aIPL encodes abstract representations of action effect structures independently of motion and object identity" (lines 127-128, Introduction) do not seem fully justified when action-PLD cross-decoding is overall stronger than action-animation cross-decoding in aIPL. Is the claim, then, that in addition to being sensitive to motion patterns, aIPL contains a neural code for abstracted causal effects, e.g. involving a separate neural subpopulation or a different coding scheme. Moreover, if sensitivity to motion patterns is not specific to SPL, but can be found in a broad network of areas (including aIPL itself), can it really be claimed that this area plays a specific role, similar to the specific role of aIPL in encoding causal effects? There is indeed, as can be seen in Figure 3A, a difference between action-PLD decoding in SPL and aIPL, but based on the searchlight map shown in Figure 3B I would guess that a similar difference would be found by comparing aIPL to several other regions. The authors should clarify these ambiguities.

We thank the reviewer for this careful assessment. The observation of action-PLD cross-decoding in aIPL is indeed not straightforward to interpret: It could mean that aIPL encodes both body movements and action effect structures by different neural subpopulations. Or it could mean that representations of action effect structures were also activated by the PLDs, which lead to successful decoding in the action-PLD cross-decoding. Our revision allows a more nuanced view on this issue:

First, we included the results of a behavioral test show that PLDs at least weakly allow for recognition of the specific actions (see our response to the second comment), which in turn might activate action effect structure representations. Second, the finding that also the cross-decoding between animations and PLDs revealed effects in left and right aIPL (as pointed out by the reviewer in the second comment) supports the interpretation that PLDs have activated, to some extent, action effect structure representations.

On the other hand, if aIPL encodes only action-effect-structures, that were also captured in the action-PLD cross-decoding, we would expect that the RDMs in aIPL are similar for the action-PLD and action-animation cross-decoding. However, the cluster analysis (see our response to Reviewer 1 above) does not show this; rather, all action-PLD RDMs are representationally more similar with each other than with action-animation RDMs, specifically with regard to aIPL. In addition, the RSA revealed sensitivity to manuality and kinematics also in aIPL. This suggests that the action-PLD decoding in aIPL was at least partially driven by representations related to body movements.

Taken together, these findings suggest that aIPL encodes also body movements. In fact, we didn't want to make the strong claim that aIPL is selectively representing action effect structures. Rather, we think that our results show that aIPL and SPL are disproportionally sensitive to action effects and body movements, respectively. We added this in our revised discussion:

"The action-PLD cross-decoding revealed widespread effects in LOTC and parietal cortex, including aIPL. What type of representation drove the decoding in aIPL? One possible interpretation is that aIPL encodes both body movements (isolated by the action-PLD cross-decoding) and action effect structures (isolated by the action-animation cross-decoding). Alternatively, aIPL selectively encodes action effect structures, which have been activated by the PLDs. A behavioral test showed that PLDs at least weakly allow for recognition of the specific actions (Tab. S2), which might have activated corresponding action effect structure representations. In addition, the finding that aIPL revealed effects for the cross-decoding between animations and PLDs further supports the interpretation that PLDs have activated, at least to some extent, action effect structure representations. On the other hand, if aIPL encodes only action effect structures, we would expect that the representational similarity patterns in aIPL are similar for the action-PLD and action-animation cross-decoding. However, this was not the case; rather, the representational similarity pattern in aIPL was more similar to SPL for the action-PLD decoding, which argues against distinct representational content in aIPL vs. SPL isolated by the action-PLD decoding. In addition, the RSA revealed sensitivity to manuality and kinematics also in aIPL, which suggests that the action-PLD decoding in aIPL was at least partially driven by representations related to body movements. Taken together, these findings suggest that aIPL encodes not only action effect structures, but also representations related to body movements. Likewise, also SPL shows some sensitivity to action effect structures, as demonstrated by effects in SPL for the action-animation and pantomime-animation cross-decoding. Thus, our results suggest that aIPL and SPL are not selectively but disproportionally sensitive to action effects and body movements, respectively."

A clarification to the sentence "aIPL encodes abstract representations of action effect structures independently of motion and object identity": Here we are referring to the action-animation cross decoding only; specifically, the fact that because the animations did not show body motion and concrete objects, the representations isolated in the action-animation cross decoding must be independent of body motion and concrete objects. This does not rule out that the same region encodes other kinds of representations in addition.

And another side note to the RSA: It might be tempting to test the "effects" model (distinguishing change of shape, change of location and ingest) also in the action-PLD multiple regression RSA in order to test whether this model explains additional variance in aIPL, which would point towards action effect structure representations. However, the "effect type" model is relatively strongly correlated with the "manuality" model (VIF=4.2), indicating that multicollinearity might exist. We therefore decided to not include this model in the RSA. However, we nonetheless tested the inclusion of this model and did not find clear effects for the "effects" model in aIPL (but in LOTC). The other models revealed largely similar effects as the RSA without the "effects" model, but the effects appeared overall noisier. In general, we would like to emphasize that an RSA with just 5 actions is not ideal because of the small number of pairwise comparisons, which increases the chance for coincidental similarities between model and neural RDMs. We therefore marked this analysis as "exploratory" in the article.

(2) Causal effect information in PLDs: the reasoning behind the use of PLD stimuli is to have a condition that isolates motion patterns from the causal effects of actions. However, it is not clear whether PLDs really contain as little information about action effects as claimed. Cross-decoding between animations and PLDs is significant in both aIPL and LOTC, as shown in Figure 4. This indicates that PLDs do contain some information about action effects. This could also be tested behaviorally by asking participants to assign PLDs to the correct action category. In general, disentangling the roles of motion patterns and implied causal effects in driving action-PLD cross-decoding (which is the main dependent variable in the paper) would strengthen the paper's message. For example, it is possible that the strong action-PLD cross-decoding observed in aIPL relies on a substantially different encoding from, say, SPL, an encoding that perhaps reflects causal effects more than motion patterns. One way to exploratively assess this would be to integrate the clustering analysis shown in Figure S1 with a more complete picture, including animation-PLD and action-PLD decoding in aIPL.

With regard to the suggestion to behaviorally test how well participants can grasp the underlying action effect structures: We indeed did a behavioral experiment to assess the recognizability of actions in the PLD stick figures (as well as in the pantomimes). In short, this experiment revealed that participants could not well recognize the actions in the PLD stick figures and often confused them with kinematically similar but conceptually different actions (e.g. breaking  shaking, hitting  swiping, squashing  knitting). However, the results also show that it was not possible to completely eliminate that PLDs contain some information about action effects.

Because we considered this behavioral experiment as a standard assessment of the quality of the stimuli, we did not report them in the original manuscript. We now added an additional section to the methods that describes the behavioral experiments in detail:

"To assess how much the animations, PLD stick figures, and pantomimes were associated with the specific action meanings of the naturalistic actions, we performed a behavioral experiment. 14 participants observed videos of the animations, PLDs (without stick figures), and pantomimes in three separate sessions (in that order) and were asked to describe what kind of actions the animations depict and give confidence ratings on a Likert scale from 1 (not confident at all) to 10 (very confident). Because the results for PLDs were unsatisfying (several participants did not recognize human motion in the PLDs), we added stick figures to the PLDs as described above and repeated the rating for PLD stick figures with 7 new participants, as reported below.

A general observation was that almost no participant used verb-noun phrases (e.g. "breaking a stick") in their descriptions for all stimulus types. For the animations, the participants used more abstract verbs or nouns to describe the actions (e.g. dividing, splitting, division; Tab. S1). These abstract descriptions matched the intended action structures quite well, and participants were relatively confident about their responses (mean confidences between 6 and 7.8). These results suggest that the animations were not substantially associated with specific action meanings (e.g. "breaking a stick") but captured the coarse action structures. For the PLD stick figures (Tab. S2), responses were more variable and actions were often confused with kinematically similar but conceptually different actions (e.g. breaking  shaking, hitting  turning page, squashing  knitting). Confidence ratings were relatively low (mean confidences between 3 and 5.1). These results suggest that PLD stick figures, too, were not substantially associated with specific action meanings and additionally did not clearly reveal the underlying action effect structures. Finally, pantomimes were recognized much better, which was also reflected in high confidence ratings (mean confidences between 8 and 9.2; Tab. S3). This suggests that, unlike PLD stick figures, pantomimes allowed much better to access the underlying action effect structures."

We also agree with the second suggestion to investigate in more detail the representational profiles in aIPL and SPL. We think that the best way to do so is the RSA that we reported above. However, to provide a complete picture of the results, we also added the whole brain maps and RDMs for the animation-pantomime, animation-PLD, pantomime-PLD, and action-pantomime to the supplementary information.

(3) Nature of the motion representations: it is not clear what the nature of the putatively motion-driven representation driving action-PLD cross-decoding is. While, as you note in the Introduction, other regions such as the superior temporal sulcus have been extensively studied, with the understanding that they are part of a feedforward network of areas analyzing increasingly complex motion patterns (e.g. Riese & Poggio, Nature Reviews Neuroscience 2003), it doesn't seem like the way in which SPL represents these stimuli are similarly well-understood. While the action-PLD cross-decoding shown here is a convincing additional piece of evidence for a motion-based representation in SPL, an interesting additional analysis would be to compare, for example, RDMs of different actions in this region with explicit computational models. These could be, for example, classic motion energy models inspired by the response characteristics of regions such as V5/MT, which have been shown to predict cortical responses and psychophysical performance both for natural videos (e.g. Nishimoto et al., Current Biology 2011) and PLDs (Casile & Giese Journal of Vision 2005). A similar cross-decoding analysis between videos and PLDs as that conducted on the fMRI patterns could be done on these models' features, obtaining RDMs that could directly be compared with those from SPL. This would be a very informative analysis that could enrich our knowledge of a relatively unexplored region in action recognition. Please note, however, that action recognition is not my field of expertise, so it is possible that there are practical difficulties in conducting such an analysis that I am not aware of. In this case, I kindly ask the authors to explain what these difficulties could be.

Thank you for this very interesting suggestion. We conducted a cross-decoding analysis that was based on the features of motion energy models as described in Nishimoto et al. (2011). Control analyses within each stimulus type revealed high decoding accuracies (animations: 100%, PLDs: 100%, pantomimes: 65%, actions: 55%), which suggests that the motion energy data generally contains information that can be detected by a classifier. However, the cross-decoding between actions and PLDs was at chance (20%), and the classification matrix did not resemble the neural RDMs. We also tested optical flow vectors as input to the decoding, which revealed similarly high decoding for the within-stimulus-type decoding (animations: 75%, PLDs: 100%, pantomimes: 65%, actions: 40%), but again at-chance decoding for action-PLD (20%), notably with a very different classification pattern:

**Author response image 1. sa2fig1:** 

Given these mixed results, we decided not to use these models for a statistical comparison with the neural action-PLD RDMs.

It is notable that the cross-decoding worked generally less well for decoding schemes that involve PLDs, which is likely due to highly different feature complexity of actions and PLDs: Naturalistic actions have much richer visual details, texture, and more complex motion cues. Therefore, motion energy features extracted from these videos likely capture a mixture of both fine-grained and broad motion information across different spatial frequencies. By contrast, motion energy features of PLDs are sparse and might not match the features of naturalistic actions. In a way, this was intended, as we were interested in higher-level body kinematics rather than lower-level motion features. We therefore decided to use a different approach to investigate the representational structure found in the action-PLD cross-decoding: As the PLDs were based on kinematic recordings of actions that were carried out in exactly the same manner as the naturalistic actions, we computed the dissimilarity of the 5 actions based on the kinematic marker positions. Specifically, we averaged the kinematic data across the 2 exemplars per PLD, vectorized the 3D marker positions of all time points of the PLDs (3 dimensions x 13 markers x 200 time points), computed the pairwise correlations between the 5 vectors, and converted the correlations into dissimilarity values by subtracting 1 - r. This RDM was then compared with the neural RDMs extracted from the action-PLD cross-decoding. This was done using a multiple regression RSA (see also our response to Reviewer 1's public comment 2), which allowed us to statistically test the kinematic model against other dissimilarity models: a categorical model of manuality (uni- vs. bimanual) and an action-specific model that discriminates each specific action from each other with equal distance.

This analysis revealed interesting results: the kinematic model explained the representational variance in bilateral SPL and (particularly right) pSTS as well as in right fusiform cortex and early visual cortex. The action-specific model revealed effects restricted to bilateral LOTC. The manuality model revealed widespread effects throughout the action observation network but not in EVC.

(4) Clustering analysis: I found the clustering analysis shown in Figure S1 very clever and informative. However, there are two things that I think the authors should clarify. First, it's not clear whether the three categories of object change were inferred post-hoc from the data or determined beforehand. It is completely fine if these were just inferred post-hoc, I just believe this ambiguity should be clarified explicitly. Second, while action-anim decoding in aIPL and LOTC looks like it is consistently clustered, the clustering of action-PLD decoding in SPL and LOTC looks less reliable. The authors interpret this clustering as corresponding to the manual vs. bimanual distinction, but for example "drink" (a unimanual action) is grouped with "break" and "squash" (bimanual actions) in left SPL and grouped entirely separately from the unimanual and bimanual clusters in left LOTC. Statistically testing the robustness of these clusters would help clarify whether it is the case that action-PLD in SPL and LOTC has no semantically interpretable organizing principle, as might be the case for a representation based entirely on motion pattern, or rather that it is a different organizing principle from action-anim, such as the manual vs. bimanual distinction proposed by the authors. I don't have much experience with statistical testing of clustering analyses, but I think a permutation-based approach, wherein a measure of cluster robustness, such as the Silhouette score, is computed for the clusters found in the data and compared to a null distribution of such measures obtained by permuting the data labels, should be feasible. In a quick literature search, I have found several papers describing similar approaches: e.g. Hennig (2007), "Cluster-wise assessment of cluster stability"; Tibshirani et al. (2001) "Estimating the Number of Clusters in a Data Set Via the Gap Statistic". These are just pointers to potentially useful approaches, the authors are much better qualified to pick the most appropriate and convenient method. However, I do think such a statistical test would strengthen the clustering analysis shown here. With this statistical test, and the more exhaustive exposition of results I suggested in point 2 above (e.g. including animation-PLD and action-PLD decoding in aIPL), I believe the clustering analysis could even be moved to the main text and occupy a more prominent position in the paper.

With regard to the first point, we clarified in the methods that we inferred the 3 broad action effect categories after the stimulus selection: "This categorization was not planned before designing the study but resulted from the stimulus selection."

Thank you for your suggestion to test more specifically the representational organization in the action-PLD and action-animation RDMs. However, after a careful assessment, we decided to replace the cluster analysis with an RSA. We did this for two reasons:

First, we think that RSA is a better (and more conventional) approach to statistically investigate the representational structure in the ROIs (and in the whole brain). The RSA allowed us, for example, to specifically test the mentioned distinction between unimanual and bimanual actions, and to test it against other models, i.e., a kinematic model and an action-specific model. This indeed revealed interesting distinct representational profiles of SPL and LOTC.

Second, we learned that the small number of items (5) is generally not ideal for cluster analyses (absolute minimum for meaningful interpretability is 4, but to form at least 2-3 clusters a minimum of 10-15 items is usually recommended). A similar rule of thumb applies to methods to statistically assess the reliability of cluster solutions (e.g., Silhouette Scores, Cophenetic Correlation Coefficient, Jaccard Coefficient). Finally, the small number of items is not ideal to run a permutation test because the number of unique permutations (for shuffling the data labels: 5! = 30) is insufficient to generate a meaningful null distribution. We therefore think it is best to discard the cluster analysis altogether. We hope you agree with this decision.

(5) ROI selection: this is a minor point, related to the method used for assigning voxels to a specific ROI. In the description in the Methods (page 16, lines 514-24), the authors mention using the MNI coordinates of the center locations of Brodmann areas. Does this mean that then they extracted a sphere around this location, or did they use a mask based on the entire Brodmann area? The latter approach is what I'm most familiar with, so if the authors chose to use a sphere instead, could they clarify why? Or, if they did use the entire Brodmann area as a mask, and not just its center coordinates, this should be made clearer in the text.

We indeed used a sphere around the center coordinate of the Brodmann areas. This was done to keep the ROI sizes / number of voxels constant across ROIs. Since we aimed at comparing the decoding accuracies between aIPL and SPL, we thereby minimized the possibility that differences in decoding accuracy between ROIs are due to ROI size differences. The approach of using spherical ROIs is a quite well established practice that we are using in our lab by default (e.g. Wurm & Caramazza, NatComm, 2019; Wurm & Caramazza, NeuroImage, 2019; Karakose, Caramazza, & Wurm, NatComm, 2023). We clarified that we used spherical ROIs to keep the ROI sizes constant in the revised manuscript.

**Reviewer #3 (Public Review):**
This study tests for dissociable neural representations of an observed action's kinematics vs. its physical effect in the world. Overall, it is a thoughtfully conducted study that convincingly shows that representations of action effects are more prominent in the anterior inferior parietal lobe (aIPL) than the superior parietal lobe (SPL), and vice versa for the representation of the observed body movement itself. The findings make a fundamental contribution to our understanding of the neural mechanisms of goal-directed action recognition, but there are a couple of caveats to the interpretation of the results that are worth noting:(1) Both a strength of this study and ultimately a challenge for its interpretation is the fact that the animations are so different in their visual content than the other three categories of stimuli. On one hand, as highlighted in the paper, it allows for a test of action effects that is independent of specific motion patterns and object identities. On the other hand, the consequence is also that Action-PLD cross-decoding is generally better than Action-Anim cross-decoding across the board (Figure 3A) - not surprising because the spatiotemporal structure is quite different between the actions and the animations. This pattern of results makes it difficult to interpret a direct comparison of the two conditions within a given ROI. For example, it would have strengthened the argument of the paper to show that Action-Anim decoding was better than Action-PLD decoding in aIPL; this result was not obtained, but that could simply be because the Action and PLD conditions are more visually similar to each other in a number of ways that influence decoding. Still, looking WITHIN each of the Action-Anim and Action-PLD conditions yields clear evidence for the main conclusion of the study.

The reviewer is absolutely right: Because the PLDs are more similar to the actions than the animations, a comparison of the effects of the two decoding schemes is not informative. As we also clarified in our response to Reviewer 2, we cannot rule out that the action-PLD decoding picked up information related to action effect structures. Thus, the only firm conclusion that we can draw from our study is that aIPL and SPL are disproportionally sensitive to action effects and body movements, respectively. We clarified this point in our revised discussion.

(2) The second set of analyses in the paper, shown in Figure 4, follows from the notion that inferring action effects from body movements alone (i.e., when the object is unseen) is easier via pantomimes than with PLD stick figures. That makes sense, but it doesn't necessarily imply that the richness of the inferred action effect is the only or main difference between these conditions. There is more visual information overall in the pantomime case. So, although it's likely true that observers can more vividly infer action effects from pantomimes vs stick figures, it's not a given that contrasting these two conditions is an effective way to isolate inferred action effects. The results in Figure 4 are therefore intriguing but do not unequivocally establish that aIPL is representing inferred rather than observed action effects.

We agree that higher decoding accuracies for Action-Pant vs. Action-PLD and Pant-PLD could also be due to visual details (in particular of hands and body) that are more similar in actions and pantomimes relative to PLDs. However, please note that for this reason we included also the comparison of Anim-Pant vs. Anim-PLD. For this comparison, visual details should not influence the decoding. We clarified this point in our revision.

**Recommendations for the authors:**

**Reviewer #1 (Recommendations For The Authors):**
It struck me that there are structural distinctions amongst the 5 action kinds that were not highlighted and may have been unintentional. Specifically, three of the actions are "unary" in a sense: break(object), squash(object), hit(object). One is "binary": place(object, surface), and the fifth (drink) is perhaps ternary - transfer(liquid, cup, mouth)? Might these distinctions be important for the organization of action effects (or actions generally)?

This is an interesting aspect that we did not think of yet. We agree that for the organization of actions (and perhaps action effects) this distinction might be relevant. One issue we noticed, however, is that for the animations the suggested organization might be less clear, in particular for "drink" as ternary, and perhaps also for "place" as binary. Thus, in the action-animation cross-decoding, this distinction - if it exists in the brain - might be harder to capture. We nonetheless tested this distinction. Specifically, we constructed a dissimilarity model (using the proposed organization, valency model hereafter) and tested it in a multiple regression RSA against an effect type model and two other models for specific actions (discriminating each action from each other with the same distance) and motion energy (as a visual control model). This analysis revealed no effects for the "valency" model in the ROI-based RSA. Also a searchlight analysis revealed no effects for this model. Since we think that the valency model is not ideally suited to test representations of action effects (using data from the action-animation cross-decoding) and to make the description of the RSA not unnecessarily complicated, we decided to not include this model in the final RSA reported in the manuscript.

In general, I found it surprising that the authors treated their LOTC findings as surprising or unexpected. Given the long literature associating this region with several high-level visual functions related to body perception, action perception, and action execution, I thought there were plenty of a priori reasons to investigate the LOTC's behaviour in this study. Looking at the supplementary materials, indeed some of the strongest effects seem to be in that region.

(Likewise, classically, the posterior superior temporal sulcus is strongly associated with the perception of others' body movements; why not also examine this region of interest?)

One control analysis that would considerably add to the strength of the authors' conclusions would be to examine how actions could be cross-decoded (or not) in the early visual cortex. Especially in comparisons of, for example, pantomime to full-cue video, we might expect a high degree of decoding accuracy, which might influence the way we interpret similar decoding in other "higher level" regions.

We agree that it makes sense to also look into LOTC and pSTS, and also EVC. We therefore added ROIs for these regions: For EVC and LOTC we used the same approach based on Brodmann areas as for aIPL and SPL, i.e., we used BA 17 for V1 and BA 19 for LOTC. For pSTS, we defined the ROI based on a meta analysis contrast for human vs. non-human body movements (Grobras et al., HBM 2012). Indeed we find that the strongest effects (for both action effect structures and body movements) can be found in LOTC. We also found effects in EVC that, at least for the action-animation cross-decoding, are more difficult to interpret. To test for a coincidental visual confound between actions and animations, we included a control model for motion energy in the multiple regression RSA, which could indeed explain some of the representational content in V1. However, also the effect type model revealed effects in V1, suggesting that there were additional visual features that caused the action-animation cross-decoding in V1. Notably, as pointed out in our response to the Public comments, the representational organization in V1 was relatively distinct from the representational organization in aIPL and LOTC, which argues against the interpretation that effects in aIPL and LOTC were driven by the same (visual) features as in V1.

Regarding the analyses reported in Figure 4: wouldn't it be important to also report similar tests for SPL?

In the analysis of implied action effect structures, we focused on the brain regions that revealed robust effects for action-animation decoding in the ROI and the searchlight analysis, that is, aIPL and SPL. However, we performed a whole brain conjunction analysis to search for other brain regions that show a profile for implied action effect representation. This analysis (that we forgot to mention in our original manuscript; now corrected) did not find evidence for implied action effect representations in SPL.

However, for completeness, we also added a ROI analysis for SPL. This analysis revealed a surprisingly complex pattern of results: We observed stronger decoding for Anim-Pant vs. Anim-PLD, whereas there were no differences for the comparisons of Action-Pant with Action-PLD and Pant-PLD:

This pattern of results is not straightforward to explain: First, the equally strong decoding for Action-Pant, Action-PLD, and Pant-PLD suggests that SPL is not substantially sensitive to body part details. Rather, the decoding relied on the coarse body part movements, independently of the specific stimulus type (action, pantomime, PLD). However, the stronger difference between Anim-Pant and Anim-PLD suggests that SPL is also sensitive to implied AES. This appears unlikely, because no effects (in left aIPL) or only weak effects (in right SPL) were found for the more canonical Action-Anim cross-decoding. The Anim-Pant cross-decoding was even stronger than the Action-Anim cross-decoding, which is counterintuitive because naturalistic actions contain more information than pantomimes, specifically with regard to action effect structures. How can this pattern of results be interpreted? Perhaps, for pantomimes and animations, not only aIPL and LOTC but also SPL is involved in inferring (implied) action effect structures. However, for this conclusion, also differences for the comparison of Action-Pant with Action-PLD and for Action-Pant with Pant-PLD should be found. Another non-mutually exclusive interpretation is that both animations and pantomimes are more ambiguous in terms of the specific action, as opposed to naturalistic actions. For example, the squashing animation and pantomime are both ambiguous in terms of what is squashed/compressed, which might require additional load to infer both the action and the induced effect. The increased activation of action-related information might in turn increase the chance for a match between neural activation patterns of animations and pantomimes.

In any case, these additional results in SPL do not question the effects reported in the main text, that is, disproportionate sensitivity for action effect structures in right aIPL and LOTC and for body movements in SPL and other AON regions. The evidence for implied action effect structures representation in SPL is mixed and should be interpreted with caution.

We added this analysis and discussion as supplementary information.

Statistical arguments that rely on "but not" are not very strong, e.g. "We found higher cross-decoding for animation-pantomime vs. animation-PLD in right aIPL and bilateral LOTC (all t(23) > 3.09, all p < 0.0025; one-tailed), but not in left aIPL (t(23) = 0.73, p = 0.23, one-tailed)." Without a direct statistical test between regions, it's not really possible to support a claim that they have different response profiles.

Absolutely correct. Notably, we did not make claims about different profiles of the tested ROIs with regard to implied action effect representations. But of course it make sense to test for differential profiles of left vs. right aIPL, so we have added a repeated measures ANOVA to test for an interaction between TEST (animation-pantomime, animation-PLD) and ROI (left aIPL, right aIPL), which, however, was not significant (F(1,23)=3.66, p = 0.068). We included this analysis in the revised manuscript.

**Reviewer #2 (Recommendations for The Authors):**
(1) I haven't found any information about data and code availability in the paper: is the plan to release them upon publication? This should be made clear.

Stimuli, MRI data, and code are deposited at the Open Science Framework (https://osf.io/am346/). We included this information in the revised manuscript.

(2) Samples of videos of the stimuli (or even the full set) would be very informative for the reader to know exactly what participants were looking at.

We have uploaded the full set of stimuli on OSF (https://osf.io/am346/).

(3) Throughout the paper, decoding accuracies are averaged across decoding directions (A->B and B->A). To my knowledge, this approach was proposed in van den Hurk & Op de Beeck (2019), "Generalization asymmetry in multivariate cross-classification: When representation A generalizes better to representation B than B to A". I believe it would be fair to cite this paper.

Absolutely, thank you very much for the hint. We included this reference in our revised manuscript.

(4) Page 3, line 70: this is a very nitpicky point, but "This suggests that body movements and the effects they induce are at least partially processed independently from each other." is a bit of an inferential leap from "these are distinct aspects of real-world actions" to "then they should be processed independently in the brain". The fact that a distinction exists in the world is a prerequisite for this distinction existing in the brain in terms of functional specialization, but it's not in itself a reason to believe that functional specialization exists. It is a reason to hypothesize that the specialization might exist and to test that hypothesis. So I think this sentence should be rephrased as "This suggests that body movements and the effects they induce might be at least partially processed independently from each other.", or something to that effect.

Your reasoning is absolutely correct. We revised the sentence following your suggestion.

(5) Page 7, line 182: the text says "stronger decoding for action-animation vs. action-PLD" (main effect of TEST), which is the opposite of what can be seen in the figure. I assume this is a typo?

Thanks for spotting this, it was indeed a typo. We corrected it: “…stronger decoding for action-PLD vs. action-animation cross-decoding..”

(6) Page 7, Figure 3B: since the searchlight analysis is used to corroborate the distinction between aIPL and SPL, it would be useful to overlay the contours of these ROIs (and perhaps LOTC as well) on the brain maps.

We found that overlaying the contours of the ROIs onto the decoding searchlight maps would make the figure too busy, and the contours would partially hide effects. However, we added a brain map with all ROIs in the supplementary information.

(7) Page 9, Figure 4A: since the distinction between the significant difference between anim-pant and anim-PLD is quite relevant in the text, I believe highlighting the lack of difference between the two decoding schemes in left aIPL (for example, by writing "ns") in the figure would help guide the reader to see the relevant information. It is generally quite hard to notice the absence of something.

We added “n.s.” to the left aIPL in Fig. 4A.

(8) Page 11, line 300: "Left aIPL appears to be more sensitive to the type of interaction between entities, e.g. how a body part or an object exerts a force onto a target object" since the distinction between this and the effect induced by that interaction" is quite nuanced, I believe a concrete example would clarify this for the reader: e.g. I guess the former would involve a representation of the contact between hand and object when an object is pushed, while the latter would represent only the object's displacement following the push?

Thank you for the suggestion. We added a concrete example: “Left aIPL appears to be more sensitive to the type of interaction between entities, that is, how a body part or an object exerts a force onto a target object (e.g. how a hand makes contact with an object to push it), whereas right aIPL appears to be more sensitive to the effect induced by that interaction (the displacement of the object following the push).”

(9) Page 12, line 376: "Informed consent, and consent to publish, was obtained from the participant in Figure 2." What does this refer to? Was the person shown in the figure both a participant in the study and an actor in the stimulus videos? Since this is in the section about participants in the experiment, it sounds like all participants also appeared in the videos, which I guess is not the case. This ambiguity should be clarified.

Right, the statement sounds misleading in the “Participants” section. We rephrased it and moved it to the “Stimuli” section: “actions…were shown in 4 different formats: naturalistic actions, pantomimes, point light display (PLD) stick figures, and abstract animations (Fig. 2; informed consent, and consent to publish, was obtained from the actor shown in the figure).”

(10) Page 15, line 492: Here, "within-session analyses" are mentioned. However, these analyses are not mentioned in the text (only shown in Figure S2) and their purpose is not clarified. I imagine they were a sanity check to ensure that the stimuli within each stimulus type could be reliably distinguished. This should be explained somewhere.

We clarified the purpose of the within session decoding analyses in the methods section: "Within-session decoding analyses were performed as sanity checks to ensure that for all stimulus types, the 5 actions could be reliably decoded (Fig. S2)."

(11) Page 20, Figure S1: I recommend using the same color ranges for the two decoding schemes (action-anim and action-PLD) in A and C, to make them more directly comparable.

Ok, done.

**Reviewer #3 (Recommendations For The Authors):**
(1) When first looking at Figure 1B, I had a hard time discerning what action effect was being shown (I thought maybe it was "passing through") Figure 2 later clarified it for me, but it would be helpful to note in the caption that it depicts breaking.

Thank you for the suggestion. Done.

(2) It would be helpful to show an image of the aIPL and SPL ROIs on a brain to help orient readers - both to help them examine the whole brain cross-decoding accuracy and to aid in comparisons with other studies.

We added a brain map with all ROIs in the supplementary information.

(3) Line 181: I'm wondering if there's an error, or if I'm reading it incorrectly. The line states "Moreover, we found ANOVA main effects of TEST (F(1,24)=33.08, p=7.4E-06), indicating stronger decoding for action-animation vs. action-PLD cross-decoding..." But generally, in Figure 3A, it looks like accuracy is lower for Action-Anim than Action-PLD in both hemispheres.

You are absolutely right, thank you very much for spotting this error. We corrected the sentence: “…stronger decoding for action-PLD vs. action-animation cross-decoding..”

(4) It might be useful to devote some more space in the Introduction to clarifying the idea of action-effect structures. E.g., as I read the manuscript I found myself wondering whether there is a difference between action effect structures and physical outcomes in general... would the same result be obtained if the physical outcomes occurred without a human actor involved? This question is raised in the discussion, but it may be helpful to set the stage up front.

We clarified this point in the introduction:

In our study, we define action effects as induced by intentional agents. However, the notion of action effect structures might be generalizable to physical outcomes or object changes as such (e.g. an object's change of location or configuration, independently of whether the change is induced by an agent or not).

(5) Regarding my public comment #2, it would perhaps strengthen the argument to run the same analysis in the SPL ROIs. At least for the comparison of Anim-Pant with Anim-PLD, the prediction would be no difference, correct?

The prediction would indeed be that there is no difference for the comparison of Anim-Pant with Anim-PLD, but also for the comparison of Action-Pant with Action-PLD and for Action-Pant with Pant-PLD, there should be no difference. As explained in our response to the public comment #2, we ran a whole brain conjunction (Fig. 4B) to test for the combination of these effects and did not find SPL in this analysis. However, we did found differences for Anim-Pant vs. Anim-PLD, which is not straightforward to interpret (see our response to your public comment #2 for a discussion of this finding).